# Characterizing Spatial Distribution of Field-Scale Snowpack using Unpiloted Aerial System (UAS) Lidar and SfM Photogrammetry

Eunsang Cho[1†], Megan Verfaillie[2,3†], Jennifer M. Jacobs[2,3], Adam G. Hunsaker[3], Franklin B. Sullivan[3], Michael Palace[3,4], Cameron Wagner[2,3*]

[1]Ingram School of Engineering, Texas State University, San Marcos, TX, USA
[2]Department of Civil and Environmental Engineering, University of New Hampshire, Durham, NH, USA
[3]Earth Systems Research Center, Institute for the Study of Earth, Oceans, and Space, University of New Hampshire, Durham, NH, USA
[4]Department of Earth Sciences, University of New Hampshire, Durham, NH, USA
[*]Present address: U.S. Army Cold Regions Research and Engineering Laboratory, Hanover, NH, USA

[†]These authors contributed equally to this work.

*Correspondence to*: Eunsang Cho (eunsang.cho@txstate.edu)

**Abstract.** Uncrewed Aerial Systems (UAS) light detection and ranging (lidar) and structure-from-motion (SfM) photogrammetry have emerged as viable methods to map high-resolution snow depths (~1 m). These technologies enable a better understanding of snowpack spatial distribution and its evolution over time, advancing hydrologic and ecological applications. In this study, a series of UAS lidar/SfM snow depth maps were collected during the 2020/21 winter season in Durham, New Hampshire, USA with three objectives: (1) quantifying UAS lidar/SfM snow depth retrieval performance using in situ magnaprobe measurements, (2) conducting a quantitative comparison of lidar and SfM retrievals of shallow snow depths (< 35 cm) throughout the winter, and (3) understanding the spatial distribution of snow depth and its relationship with terrain features. Eight UAS surveys were conducted over approximately 0.35 km$^2$ including both open fields and a mixed forest. In the field, lidar had a slightly lower error than SfM compared to in situ observations with a Mean Absolute Difference (MAD) of 3.5 cm for lidar and 4.0 cm for SfM. Snow depth maps from SfM and lidar were fairly consistent in the field with differences close to 0 cm on most dates. In the forest, SfM greatly overestimated in situ snow depths compared to lidar (lidar MAD = 6.3 cm, SfM MAD = 31.4 cm). There was also no clear agreement between SfM and lidar snow depth values for individual 1 m$^2$ pixels in the forest (MAD = 55.7 cm). Using the concept of temporal stability, we found that the spatial distribution of snow depth captured by lidar was generally consistent throughout the period indicating a strong influence from static land characteristics. Considering both areas (forest and field), the spatial distribution of snow depth was primarily influenced by vegetation type while also reflecting the effects of soil variables (e.g., soil organic matter). When the field and forest areas were analysed separately, the spatial distribution was distinctly affected by slope and the shadowing effects of the forest canopy.

# 1 Introduction

Snowpacks are vital to hydrologic, climatic, and ecological processes across multiple scales (Barnett et al., 2005; Clark et al., 2011). Snowpack distribution and its temporal evolution are important to determine snowmelt runoff, infiltration, and groundwater recharge (Carroll et al., 2019; Harpold et al., 2015; Maurer and Bowling, 2014) as well as energy partitioning processes (Lawrence and Slater, 2010; Stieglitz et al., 2001; Sturm et al., 2017). Snowpacks also exert a strong control on snow-soil interactions because the insulating capacities of snowpack affect the underlying soil freeze-thaw state influencing soil respiration, nutrient retention, and carbon dynamics (Anderton et al., 2002; Schlogl et al. 2018; Cho et al., 2021; Monson et al., 2006; Sorensen et al., 2018; Reinmann and Templer, 2018; Wilson et al., 2020; Yi et al., 2015).

The spatial variability of a snowpack is a function of static and dynamic variables and fluxes over a range of spatial scales (Clark et al., 2011; Grayson et al., 2002; Mott and Lehning, 2011; Trujillo et al., 2007). Over time, spatial patterns may evolve and change, but many hydrologic patterns persist until they are modified by weather conditions. Spatial snowpack patterns and their consistency, or repeatability, play a crucial role in various applications, including operational snowmelt predictions, the downscaling of remotely sensed or model outputs, the integration of in situ observations through upscaling, the assimilation of data to enhance model simulations, and the utilization of snowpack characteristics as proxies or analogs for similar hydrological units (Pflug and Lundquist, 2020; Cho et al., 2023). They can also provide insight into underlying landscape features, biogeochemical processes, and wildlife habitats (Boelman et al., 2019; Pflug et al., 2023).

Numerous investigations have introduced various approaches to capture snow distribution patterns and their evolution across diverse climatic and topographical environments. For example, Sturm and Wagner (2010) found that snow depth patterns remain stable across years due to persistent topographic and vegetation influences in an Arctic region, highlighting the value of empirical snow distribution patterns for improving snow model accuracy. Vögeli et al. (2016) used high-resolution airborne digital sensors to refine precipitation scaling in a snow distribution model (Alpine3D), demonstrating the potential of remote sensing data to better simulate complex snow dynamics in alpine regions. Pflug et al. (2021) examined the interannual consistency of snow patterns and proposed a downscaling approach based on historical snow patterns in the California Tuolumne River Watershed, which is particularly useful for predicting snow distribution during years with limited observations. Revuelto et al. (2020) introduced a method combining in situ snow depth measurements with terrestrial laser scanner and time-lapse photography to produce temporal snow depth distribution patterns in a subalpine mountain environment, offering a transferable approach for deriving spatial snow data from limited ground observations.

Traditionally, field (approximately 100 m) or local-scale (approximately 1 m) snow features are captured through in situ observations and field campaigns (Clark et al., 2011; Trujillo et al., 2007), whereas regional or continental-scale patterns are typically observed using airborne and satellite remote sensing techniques (Lievens et al., 2022; Painter et al., 2016; Derksen et al., 2005). Airborne and satellite remote sensing methods have provided the ability to collect snowpack data over a large spatial extent, thus expanding the understanding of snow distribution (Cho et al., 2019; Lievens et al., 2022; Painter et al., 2016; Tsang et al., 2021). However, the ability to capture small-scale snow patterns, discerned through field campaigns or less

frequent, routine operational collections, is often hindered by challenges such as weather conditions, tree canopies, and site accessibility which can lead to infrequent sampling during the winter season.

Uncrewed Aerial Systems (UASs) have been used to provide spatially continuous, opportunistic snow-covered area and snow depth observations at scales between in situ and airborne and satellite remote sensing (Bühler et al., 2016; De Michele et al., 2016; Harder et al., 2016; 2020; Meyer et al., 2022; Revuelto et al., 2021a; Geissler et al., 2023). UAS-based remote sensing enables the acquisition of data at fine spatial resolutions, reaching scales as precise as centimeters for a designated area. UAS platforms also offer a cost-effective alternative to aerial surveys, facilitating routine monitoring of snow conditions (Gaffey et al., 2020). Hence, the capabilities of UAS platforms equipped with diverse sensors can observe snowpack properties and support analyses of field-scale physical interactions between snowpacks and land/soil characteristics (Cho et al., 2021).

UAS light detection and ranging (lidar) and structure-from-motion (SfM) photogrammetry have emerged as viable methods for mapping high-resolution snow depths (~1 m), enabling a better understanding of snowpack spatial distribution and its evolution over time at the field scale (Feng et al., 2023; Harder et al., 2019; Jacobs et al., 2021; Koutantou., 2022; Geissler et al., 2023). As the use of UAS-based high-resolution snow depth mapping becomes more prevalent, there is a growing need for a comprehensive understanding of their strengths and weaknesses for capturing snowpack evolution throughout the entire snow period for various landscape features. However, investigating the transition periods between snow-on and snow-off poses challenges, primarily due to the snow becoming increasingly shallow and patchy, eventually revealing bare ground. Despite these challenges, these transition periods hold significant hydrological, ecological, and energy implications (Harrison et al., 2021; Harpold et al., 2017; Grogan et al., 2020).

This study aims to achieve three main objectives using a series of UAS lidar/SfM snow depth maps over a mixed-use temperate forest landscape: (1) quantify UAS snow depth retrieval performance by comparing it with in situ measurements, (2) conduct a quantitative comparison of lidar and SfM snow depths throughout the snow period for a range of depths that reflect the specific conditions observed in our dataset (i.e., 0 to 35 cm), and (3) gain a better understanding of the spatial distribution of snow depth, its stability over time, and its relationship with physical terrain features. This paper is organized as follows: Section 2 provides an overview of the study area, including its land characteristics. Section 3 describes the datasets utilized in the study, including UAS lidar, SfM photogrammetry, and field observations, and the methods employed, such as the relative difference concept. Section 4 presents the results, with subsections detailing comparisons between UAS snow depth and in situ measurements (4.1), as well as comparisons between lidar and SfM snow depth (4.2). Additionally, spatial patterns and temporal changes in snow depth, along with relevant physical variables characterizing those snow patterns, are discussed in Sections 4.3 and 4.4. Section 5 discusses new insights derived from the comparison results and spatial patterns of snow depth, along with the limitations of this study and future perspectives. Finally, conclusions are drawn in Section 6.

## 2 Study area

This study was conducted at the University of New Hampshire Thompson Farm Research Station in southeast New Hampshire, United States (N 43.10892°, W 70.94853°, 35 m above sea level), which was chosen for its mixed hardwood forest and open field land covers (Perron et al. 2004; Burakowski et al., 2015; Jacobs et al., 2021) that are characteristic of the region (**Figure 1**). Thompson Farm has a rich history of forest ecology research and data collection. Thompson Farm has an area of 0.83 km$^2$ and little topographic relief (18 to 36 m ASL) (Perron et al., 2004). The agricultural fields are actively managed for pasture

grass. The deciduous, mixed, and coniferous forest is composed primarily of white pine (Pinus strobus), northern red oak (Quercus rubra), red maple (Acer rubrum), shagbark hickory (Carya ovata), and white oak (Quercus alba). The forest soils are classified as Hollis/Charlton very stony-fine sandy loam and well-drained; field soils are characterized as Scantic silt-loam and poorly drained (Perron et al. 2004). There are two logging access roads running north-south through the pasture and western forest section. The winter climate at Thompson Farm has a mean winter air temperature of -3.0°C and an annual snowfall of

114 cm with three weeks to over three months of days having snow cover (Burakowski and Hamilton, 2020; Johnston et al., 2024). Snow depth can range from a trace up to 94 cm and typical snow density ranges from 100 to 400 kg/m$^3$ (Burakowski and Hamilton, 2020). The snowpack at Thompson Farm is short-lived and warm, and snow climatologies from Sturm and Liston (2021) and Johnston et al. (2024) both classify the area as ephemeral. A review of existing research on the snow classes defined by Sturm and Liston (2021) and Johnston et al. (2024) determined that, despite covering large areas in the northern

hemisphere, the ephemeral snow class is largely understudied, making new research on ephemeral snowpacks valuable.

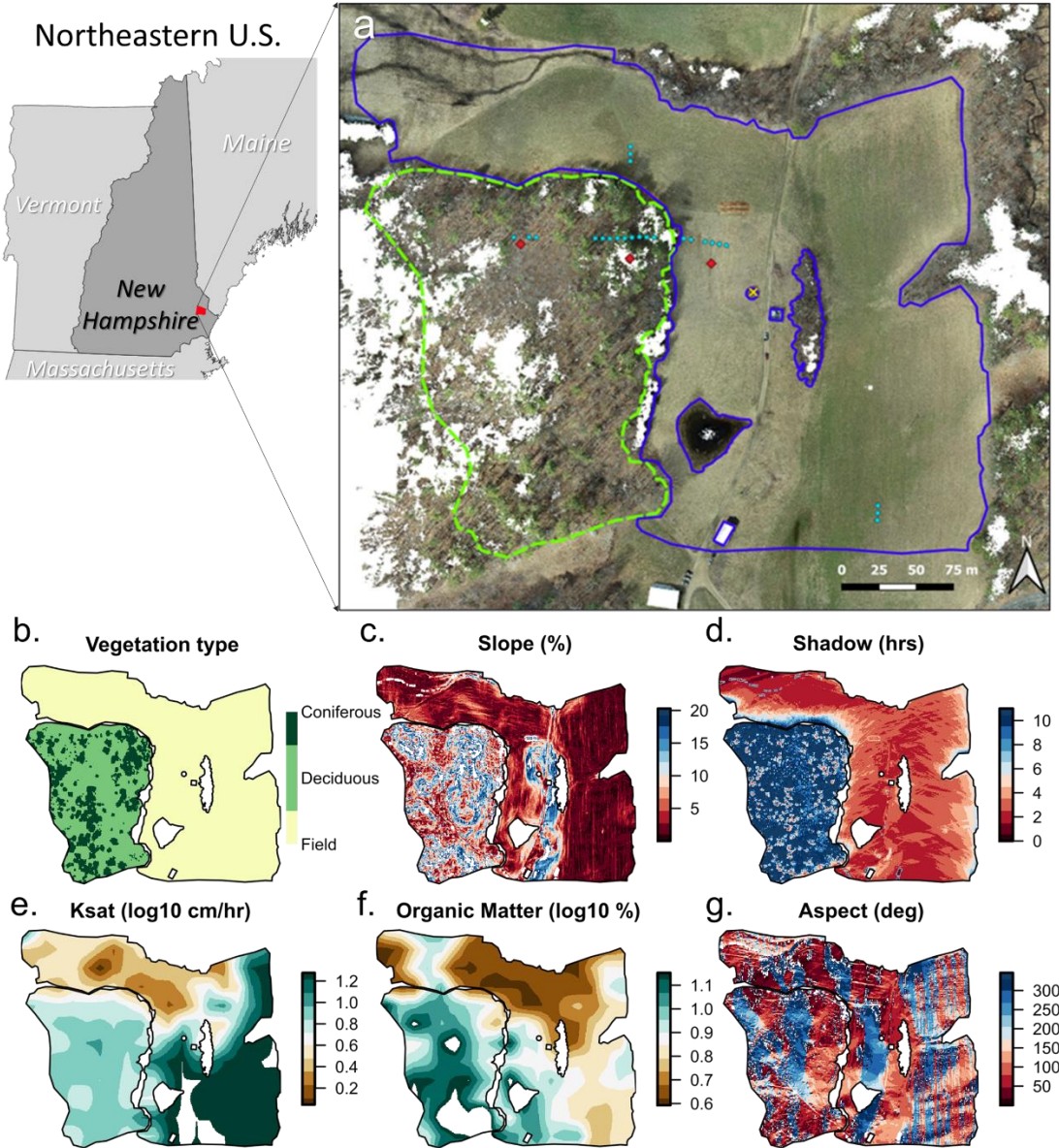

**Figure 1 Thompson Farm survey area located in Durham, NH, USA. (a) In situ sample sites and field and forest boundaries are overlain on the snow-off imagery. The pond, section of dense shrubs, outbuildings, and USCRN station in the western field were not representative of the field and were removed. Maps of (b) vegetation type, (c) slope, (d) shadow hour, (e) soil hydraulic conductivity (Ksat), (f) organic matter, and (g) aspect are shown for the field and forest areas. The derivation of each of these variables is explained in Section 3.**

## 3 Datasets and methods

A series of UAS lidar surveys, UAS SfM photogrammetry surveys, and in situ sample campaigns were conducted at Thompson Farm during the winter 2020-2021. Eight snow-on campaigns were conducted between February 10[th] and March 11[th], 2021 during which UAS lidar, UAS SfM photogrammetry and in situ data were collected (**Table S1**). The UAS snow-on surveys were conducted prior to in situ sampling on each of the campaign dates. Because compaction of underlying vegetation during snow-on periods can result in negative snow depths when compared to the snow-off baseline (Masný, Weis, and Biskupič, 2021), the snow-off baseline survey was conducted on April 2[nd], 2021 following snowmelt.

### 3.1 UAS lidar and SfM photogrammetry

The lidar sensor payload consisted of the Velodyne VLP-16 laser scanner, and the Applanix APX-15 Inertial Navigation System (INS; GNSS+IMU). The VLP-16 is a lightweight (~830 grams), low power (~8 W) sensor, which makes it ideal for UAS deployment. The sensor incorporates 16 rotating infra-red (IR) lasers that are arranged and oriented on the payload to provide a 30° along-track field of view with a cross-track field of view limited only by the range of the sensor (approximately 100 m). At an altitude of 65 m, the sensor range produces an effective cross-track field of view of approximately 98°. Each laser operates at a wavelength of 903 nm.

For these acquisition missions, the VLP-16 was hard-mounted to a DJI Matrice 600 to maintain constant lever arm offsets between the Inertial Navigation System (INS) GNSS antenna, the lidar sensor, and the INS board. As opposed to a gimbal mounted system, this hard-mounted configuration achieves a more tightly coupled system, resulting in improved point cloud geolocation accuracy. The lidar sensor was set to dual-return mode to improve ground detection in the forested areas of the field site. The system was flown at an altitude of 65 m with a flight speed of 3 m/s and ~40 m spacing between flight lines. Flights produced between ~70 –140 million returns per mission, depending on site ground conditions.

Lidar observations were georeferenced using position and attitude measurements acquired with the Applanix APX-15 INS. The INS produced 2–5 cm positional, 0.025-degree roll and pitch, and 0.08-degree true heading uncertainties following post-processing. Post-processing of INS data was performed using POSPac UAV (v. 8.2.1, Applanix Corporation 2018), correcting differentially against a permanent Continuously Operating Reference Station (CORS) at the University of New Hampshire in Durham, NH (NHUN). Position and attitude data were output as a Smoothed Best Estimate of Trajectory (SBET), then time synchronized with lidar returns to produce a georeferenced point cloud using LidarTools (v. 3.1.4, Headwall Photonics, Inc.). Three-dimensional point clouds were processed using a progressive morphological filter (PMF) within the R programing language package 'lidR' to identify ground returns. For ground classification, point clouds were chunked into 100-m square tiles with a 15-m buffer on all sides using catalogue options in lidR to ensure returns near tile edges were classified. The PMF was parameterized using a set of window sizes of 1, 3, 5, and 9 m, and elevation thresholds of 0.2, 1.5, 3, and 7 m, which were determined by varying value sets and assessing digital terrain models (DTMs) to determine the parameter sets that produced a visually smooth surface over a dense grid (Muir et al. 2017). Following ground classification for each tile, returns within the

15-m tile buffers were removed, and all resulting 100-m square ground classified tiles were merged. The result of the PMF is that non-ground returns (i.e., trees, shrubs, and noise) were filtered out of the point cloud data sets, so that only returns from ground surfaces remained. The two data sets, non-ground returns and ground returns from the original point clouds, were coded according to LAS specifications and merged. Lidar snow depths were calculated as the difference between the ground classified snow-on and snow-off elevations within each pixel. For comparison to in situ observations, the ground returns were extracted for the 1 x 1 m square sampling sites, corresponding to the alignment and orientation of the transect. The lidar snow depth was calculated as the difference between the mean snow-on and mean snow-off elevations within each sampling grid.

Photogrammetry bare-earth and snow-on elevation models were constructed from UAS-borne optical imagery. RGB images were collected with the DJI Phantom 4 Real Time Kinematic (RTK) UAS platform equipped with a 20-megapixel Complementary Metal Oxide Semiconductor (CMOS) sensor. The RTK system integrates a static base station that relays GNSS corrections to the UAS, enabling approximately 3-cm accuracy of image geotags. To ensure photogrammetry snow depth products align correctly with the UAS lidar products, the RTK base-station was placed over a monument with known coordinates which were entered into the DJI flight app. Flights were conducted at an altitude of 65 m AGL and a flight speed of 8 m/s. The shutter triggering interval was set to achieve a forward overlap of 80% between image pairs and the flight lines were spaced to achieve 80% side overlap. Three Ground Control Points (GCPs) were placed within the area of interest to verify the accuracy of the photogrammetry products. The GCPs were surveyed in using a Trimble© Geo7X GNSS Positioning Unit and Zephyr™ antenna with sub-centimeter accuracy.

The acquired image datasets were processed through the basic photogrammetry workflow using Agisoft Metashape (version 1.8.4). Sparse clouds were constructed using the default key point and tie point limits of 40,000 and 4,000, respectively. Points with high errors within the sparse clouds were then removed using the gradual selection tool. This included points exceeding the following thresholds; reprojection error > 0.5, reconstruction uncertainty > 50, and projection accuracy > 5. The camera intrinsic/extrinsic parameters were optimized following the removal of the poorly localized points. Dense clouds were produced with the quality setting set to high and depth filtering set to moderate. Ground returns were classified using the ground classification tool within Metashape. A first pass at establishing a ground surface is done by triangulating the lowest point elevation within 50-m grid cells. The default thresholds for maximum distance and angle (1 m and 15 degrees respectively) of all points relative to the triangulated surface were used to determine which points are part of the ground surface. Finally, digital elevation models (DEMs) were derived from the ground classified points within the dense clouds. Snow depth products were derived following the same procedure as the lidar by calculating the difference between the ground classified snow-on and snow-off elevations within each pixel. Additional filtering based on the point confidence metric was completed for the February 20th and 24th snow depth maps to remove the points with high uncertainty. GCPs surveyed using the base/rover equipment were used to co-register the UAS data. Linear, horizontal, and vertical shifts were applied to align all SfM and lidar DEMs to the GCPs.

## 3.2 Field observations

In situ snow depth sampling was conducted in the field and forest using two methods: a Snow-Hydro LLC magnaprobe (Sturm and Holmgren, 2018) and three Moultrie Wingscapes Birdcam Pro Field Cameras. The magnaprobe sampling followed a single long transect (18 points) and two short transects (3 points each). The long transect was approximately 145 m long and laid out from east to west (**Figure 1**). From east to west, the transect started in the open field area, then transitioned to the coniferous, then mixed, and finally, deciduous forested areas. The two short transects were located in the open field; one in the northwest portion and the other in the southeast. At each point, nine, evenly spaced measurements were taken within 1 m x 1 m grid cells. It is worth noting that some dates were missing sample points due to disturbance of the sample area, either by collection on previous days or recreational use at the site, or due to personnel and equipment limitations (**Table S1**). All sampling locations were geolocated using a Trimble Geo7X GNSS Positioning Unit and Zephyr antenna with an estimated horizontal uncertainty of 2.51 cm (standard deviation 0.95 cm) in the field and 4.17 cm (standard deviation 4.60 cm) in the forest after differential correction.

Field camera snow depths were acquired following the method used in NASA's 2020 SnowEx field campaign in Grand Mesa, CO (personal communication, 16th November 2020). The three cameras were placed in different land cover types; one in the open field, one in the coniferous forest, and one in the deciduous forest. Each camera was mounted approximately 0.85 m above the ground and placed approximately 5.5 m from its respective 1.5 meter marked PVC pole. Each PVC pole was spray-painted red and marked with 1 cm and 10 cm increments. The cameras captured images of the poles every 15-minutes for the duration of the study period. Snow depth was derived by manual inspection of the photos and recorded to the nearest cm. Precipitation equivalent and mean temperature data were measured by a NOAA Office of Oceanic and Atmospheric Research U.S. Climate Reference Network (USCRN) station (NH Durham 2 SSW) located in the western portion of the field. Hourly air temperatures at the USCRN station are averaged from two-second readings from three independent thermometers. Hourly precipitation is computed from 5-minute readings of depth change measured by a weighing precipitation gage.

## 3.3 Physical land characteristics

Land and soil characteristic variables are investigated as physical drivers of field-scale spatial distribution of snow depth. The variables used in this study are plant functional type, slope, aspect, shadow hours, saturated hydraulic conductivity ($K_{sat}$), and soil organic matter (SOM) (**Figure 1**). Mapped at a 1-m scale, all variables are derived from UAS snow-off observations except the two soil variables. The two soil variables, $K_{sat}$ and SOM, are at soil depth of 0–5 cm obtained from Probabilistic Remapping of SSURGO (POLARIS) maps at 30-m spatial resolution (Chaney et al., 2016; 2019). The soil maps were disaggregated to 1-m spatial resolution without employing interpolation methods to mitigate additional uncertainties. Vegetation cover type (field/forest) was manually delineated in geographic information system (GIS) software based on the image orthomosaics created during SfM processing. The forested area was further classified as coniferous or deciduous for the

study region by applying the Green Leaf Index (GLI) (Louhaichi, Borman, and Johnson 2001) (**Equation 1**) to the optical three-band (red, green, and blue) orthomosaics derived from the snow-off DJI Phantom 4 RTK survey.

$$\text{GLI} = \frac{(Green-Red)+(Green-Blue)}{(2*Green)+Red+Blue} \tag{1}$$

The GLI algorithm delineated the dense vegetation (conifer trees) from the less dense vegetation (leaf-off deciduous trees) (Borman and Johnson, 2001). The direct application of the GLI algorithm on the three-band orthomosaics was further filtered and refined as follows. The output was clustered using the k-means algorithm with the number of k classes equal to two: one class for coniferous trees (high GLI) and one class for deciduous trees (low GLI). Noise within the clustered GLI map was removed by convolution with a median filter. To establish continuous delineations of coniferous regions, morphological

closing was applied to the map to fill in any interior holes within the delineated regions. Forest classifications for each of the magnaprobe sample locations was estimated for a 10 m x 10 m area centered at each sampling grid based on the percent coniferous pixels (< 40% = deciduous, 40 – 60% = mixed, > 60% = coniferous). Results from the binary forest classification and the coarsened 10 m x 10 m classification are shown in **Figure S1**. The slope and aspect were derived from the UAS lidar 1 m snow-off DEM using Horn's method (Horn, 1981). The shadow hours were calculated using the unfiltered UAS LiDAR

digital terrain model and a static sun incidence angle based on the average of February 4th and March 7th. Given the minor variation in solar angles between these dates, any change in shadow hours was considered negligible for this study.

### 3.4 Relative difference concept

The relative difference concept, first introduced by Vachaud et al. (1985), has been widely used in the soil moisture remote sensing community to quantify spatio-temporal variability (or stability) of soil moisture at field or regional scales (Cho and

Choi, 2014; Cosh et al., 2004; Jacobs et al., 2004; Mohanty & Skaggs, 2001; Starks et al., 2006). In this study, we apply this concept to the UAS-lidar snow depth measurements. The relative difference in the snow depth measurements can be expressed as

$$RD_{i,t} = \frac{SND_{i,t}-spatial\,mean(SND_t)}{spatial\,mean(SND_t)} \tag{2}$$

where $SND_{i,t}$ is individual snow depth measurement at grid $i$ and date $t$, and spatial mean ($SND_t$) is the spatial mean value of

snow depth at date $t$. For each grid $i$, the mean relative difference ($MRD_i$) is the average relative difference from each of the N flights and can be calculated by

$$MRD_i = \frac{1}{N}\sum_{t=1}^{N} RD_{i,t} \tag{3}$$

# 4 Results

## 4.1 In Situ vs UAS-measured Snow Depths

Daily temperature, daily precipitation, and measured snow depths in the field and forest for the study period are shown in **Figure 2**. Between December 15th, 2020 and March 12th, 2021, the average daily temperature was -2ºC, the maximum daily temperature was 19°C on January 31st and the minimum daily temperature was -19°C on March 11th. Average wind speed was 1.4 m/s for the study period. The cumulative precipitation for the same period was 20.4 cm. The largest precipitation event was 11 cm and occurred on January 16th when temperatures were above freezing (2 – 9°C). December and early January had ephemeral snowpacks of less than 10 cm which melted within a week. A snowpack was continuously present from late January through the middle of March. The maximum snow depth measured by the cameras occurred on February 10th with 21 cm in the field, 21 cm in the deciduous forest, and 18 cm in the coniferous forest. A second peak snow depth occurred on February 20th with 20 cm in the field, 12 cm in the deciduous forest, and 19 cm in the coniferous forest. A sustained period of warm temperatures in late February and early March corresponded to a decrease in snow depth due to the warming temperatures and two rain-on-snow events. The snowpack was depleted from the entire study area by March 10th.

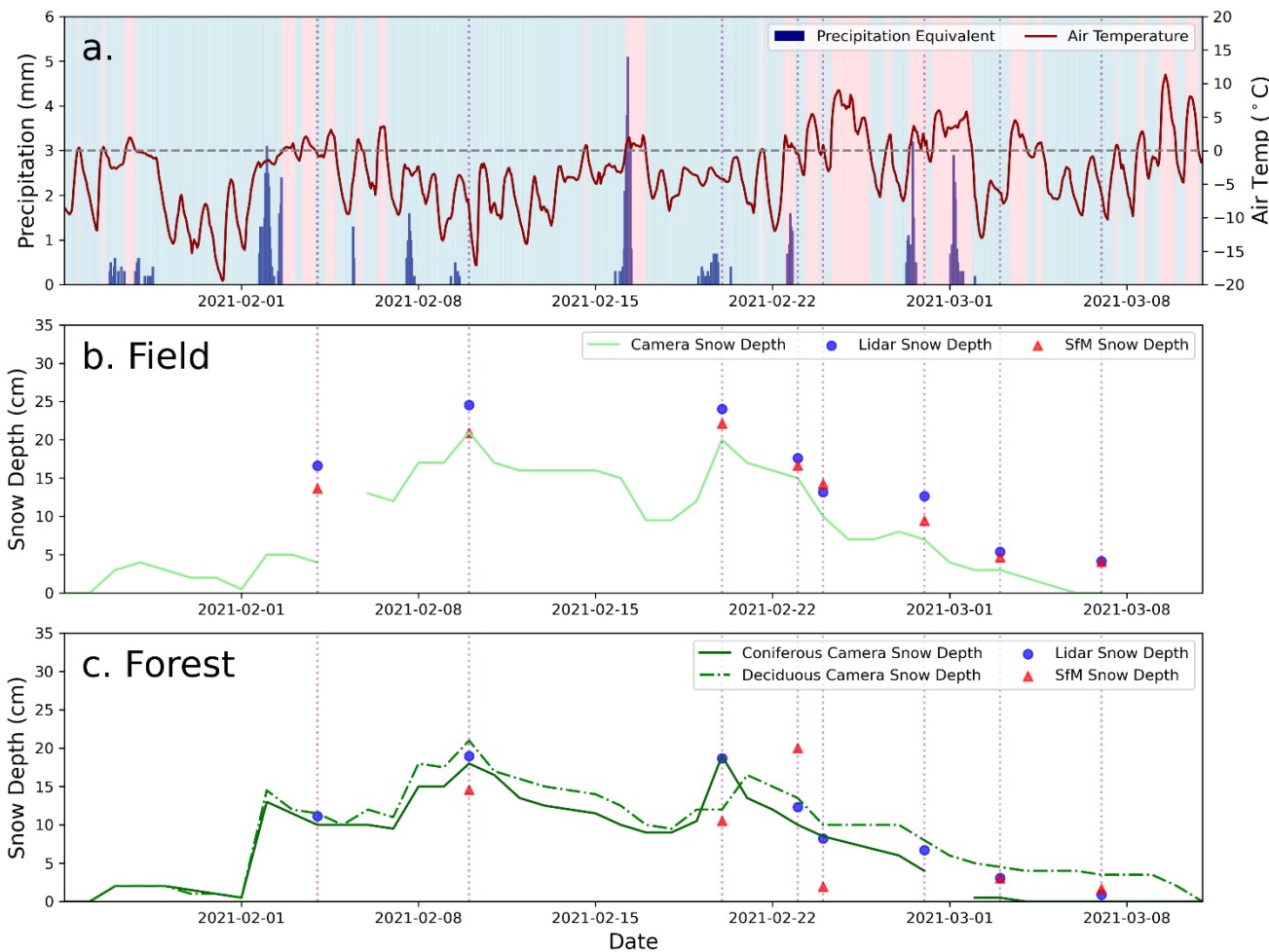

**Figure 2** Time series of conditions at the Thompson Farm, Durham NH study area during the 2021 winter season including: hourly precipitation equivalent (mm) and temperature (°C) measured by a USCRN station (a) and daily camera snow depths and median UAS-measured snow depths in the field (b) and forest (c). Dates corresponding to the in situ and UAS sampling campaigns are marked by the dotted vertical lines. Periods where the temperature was colder than 0°C are indicated by the blue plot background and periods warmer than 0°C are indicated in pink in (a). Median SfM-measured snow depths in the forest on 2/4/21 and 2/28/21 exceeded 35 cm and are not shown (172 cm and 86 cm, respectively).

**Figures 2b** and **2c** show that the eight UAS-based SfM and lidar flights captured both the snow depth peaks and the ablation period following the last peak on February 20th, referring to the phase in the seasonal snow patterns when the snowpack begins to melt and decrease in depth. During the ablation period, UAS measurements typically showed a decreasing snow depth that matched the progression from the field camera measurements. The final UAS surveys on March 7th captured the transition from a snow-cover-dominated field area on March 3rd to bare ground dominated. In the field, the UAS-based measurement techniques yielded similar snow depths throughout the winter and were able to capture snow depth changes on the order of 5 cm or less. However, the forest performance often differed between the two UAS methods. In the forest, the lidar snow depths

tracked the camera observations within 5 cm. However, the SfM method had inconsistent performance. Notably, on February 4th and February 20th, the SfM snow depths exceeded the lidar depths by more than 50 cm.

UAS-based snow depth retrievals were compared to in situ snow depths measured by the magnaprobe (**Figure 3; Table S2**).
Field camera observations were not used for validation due to their limited spatial extent compared to the magnaprobe sample
locationsFigure 3. In the field, both UAS lidar and UAS SfM snow depths were approximately 1.5 cm deeper on average than
the magnaprobe measurements. While both UAS methods tended to follow the 1:1 line, SfM had several outliers in which the
SfM snow depth overestimated the in situ measurements. Overall, the UAS lidar performance was modestly better in the open
field than the UAS SfM as compared to the magnaprobe measurements based on the Mean Absolute Difference (MAD) (SfM
= 4.0 cm, Lidar = 3.5 cm) and the fitted linear regression line $r^2$ values (SfM = 0.51, Lidar = 0.73). Samples from the field had
275 better agreement between UAS and magnaprobe measurements than the forest. In the forest, the MAD values increased
modestly for the lidar, but sharply for the SfM snow depths when compared to the magnaprobe measurements (SfM = 31.4
cm, Lidar = 6.3 cm). In addition, lidar measurements had a much higher $r^2$ value than SfM in the forest (SfM = 0.02, Lidar =
0.70). The UAS-measured snow depths in the forest are also shifted to the right of the 1:1 line, indicating that they tended to
measure shallower snow depths than the magnaprobe. This does not necessarily indicate an error in the UAV measurements
because a previous study by Proulx et al. (2023) at this study site demonstrated that the magnaprobe tends to overprobe in the
forest.

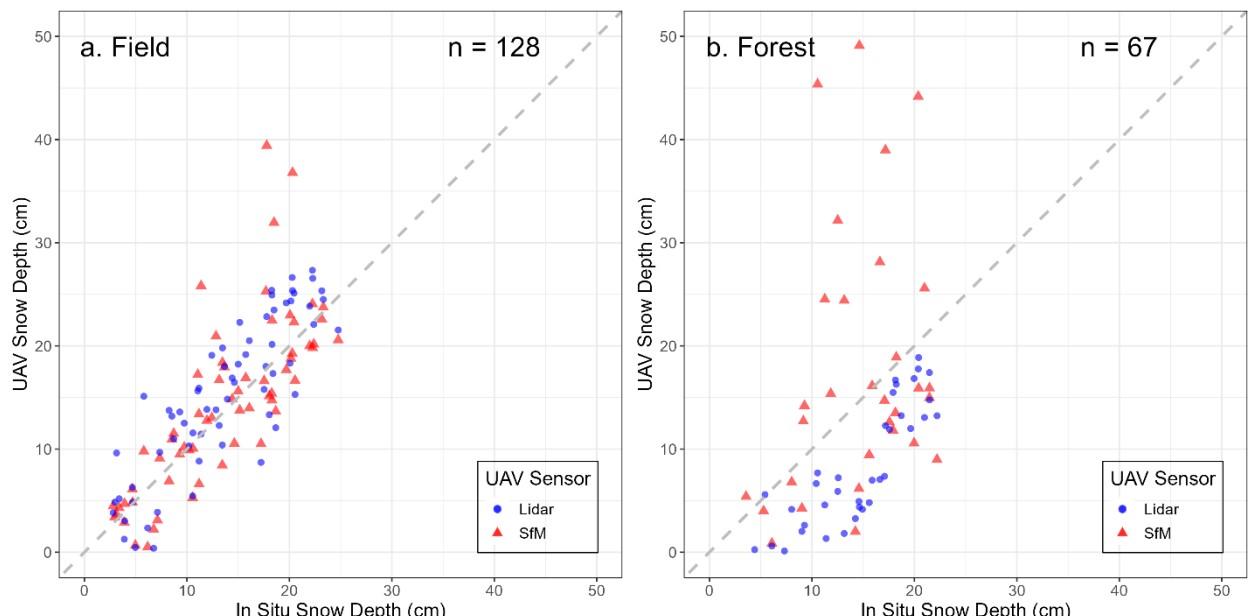

**Figure 3 UAV-based snow depth measurements compared to in situ snow depth measurements from the magnaprobe. UAS-measured snow depths are shown for pixels overlapping the magnaprobe sample locations. Measurements collected on all dates are**
285 **shown for the field (a) and forest (b). N-values indicate the number of samples shown within the plot axes. UAV-measured snow depths which exceeded 50 cm are not shown (1 SfM field, 6 SfM forest). Magnaprobe sample locations corresponding to UAS pixels with missing snow depth values are not shown (1 SfM field, 6 SfM forest, 7 lidar forest).**

## 4.2 Comparison between lidar and SfM Snow Depth

**Figure 4** shows a direct comparison between the snow depths at individual 1 m x 1 m pixels measured by lidar and SfM over the entire study area. The field was divided into three areas based on an early study (Cho et al., 2021) showing distinct topographic and soil characteristics in each section. While there is considerable scatter for the individual pixels in all field areas, the SfM and lidar snow depths tend to agree fairly well based on MAD (east = 4.4 cm, west = 2.9 cm, northwest = 4.3 cm). Compared to the northwest and east fields, the west field had the most similar snow depth values for SfM and lidar. In that field, both lidar and SfM techniques captured relatively deeper snow depths ranging from 50 to 100 cm. In the northwest and east fields, SfM snow depths are frequently much deeper than the lidar snow depths. In contrast, there was no clear agreement between SfM and lidar snow depth at the 1 m resolution in the forest (MAD = 55.7 cm), largely due to extensive regions in which SfM snow depths were anomalously high.

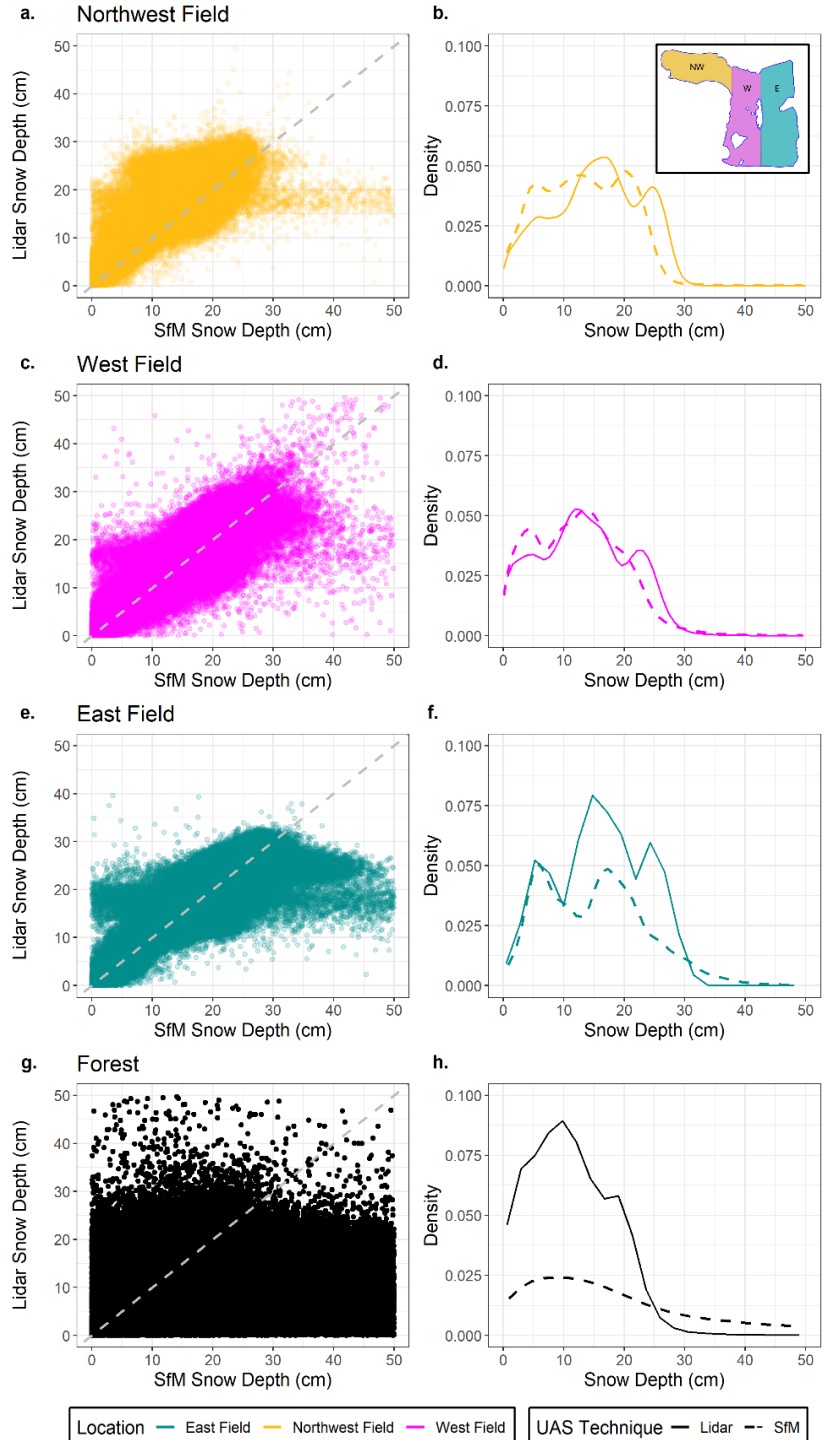

**Figure 4 Comparison of SfM and lidar measurements for all sample dates by location. Scatterplots (left) compare snow depth for the three field areas (a, c, e) and the forest (g). Probability density plots (right) show the distribution of snow depth values for the three field areas (b, d, f) and the forest (h) by UAS technique. Measured snow depths exceeding 50 cm are not shown.**

A time series of lidar and SfM snow depth maps over the entire study area are shown in **Figure 5.** While the previous section found that individual locations may have differences, these maps show that both techniques capture the differences in snow depth between flights in both the field and forest areas. The difference between the snow depth maps by sample date shows where the UAS snow depths tend to agree and disagree. The difference between SfM and lidar snow depths was fairly consistent in the field and close to 0 cm on most dates. However, on February 20th and 24th the southeastern field showed a negative difference indicating that the SfM snow depths were considerably deeper than the lidar snow depths in parts of the field. On other dates including February 4th, the SfM snow depth map was missing data from extended areas in the field. In the forest, missing or patchy SfM data occurred on many days (e.g., February 4th, 20th, and 28th). Lidar and SfM ground return point count statistics by land cover type are summarized in Tables S3 through S6. Despite the limited overlap in the forest, there is limited agreement between the two methods. Most maps show that the SfM snow depths were much deeper than the lidar snow depth through most of the forest except at the forest and field edge.

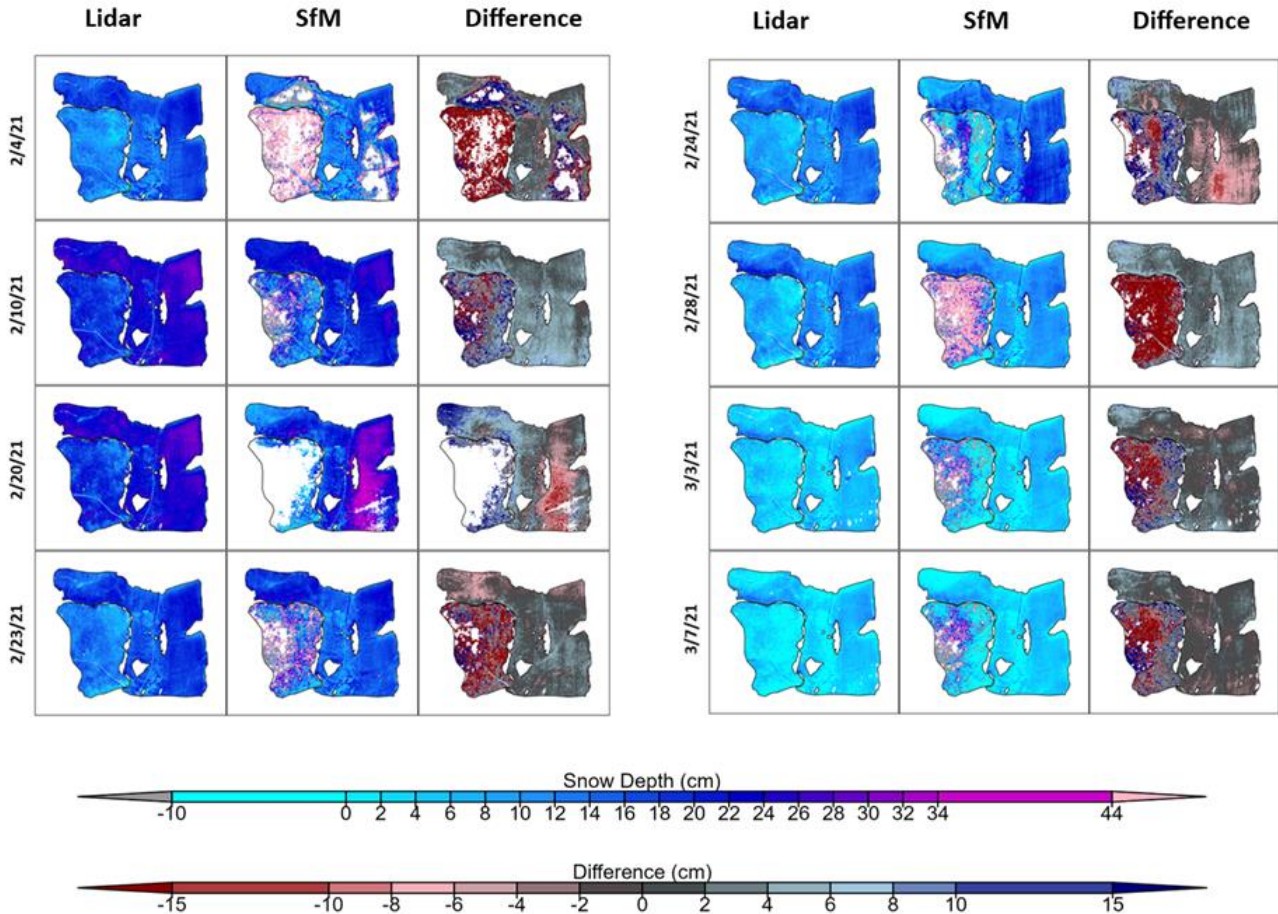

Figure 5 Time series of snow depths for UAV Lidar and SfM in the field and forest. Difference is calculated as Lidar snow depth minus SfM snow depth. All values are shown in cm.

## 4.3 Spatial Distribution of Snowpack and its Temporal Changes

To investigate the spatial distribution of snow depth over time, the MRD values were mapped (**Figure 6**). The average spatial distribution of snow depth across the study domain based on UAS lidar-derived maps from eight survey dates shows spatially distinct patterns. The field had relatively deeper snow by up to 70% greater than the spatial mean. Within the field, the snow in northern areas was generally deeper than that in the southern areas, except for near the northern edges where it was shallower. In forested areas, the snow was shallower by up to -70% relative to the spatial mean. Snow in northeastern forest areas was generally shallower than that in other forest areas. Distinct differences in the transition zones between field and forest show edge effects. Immediately south of the tree line at the northeastern extent of the field, the snowpack is noticeably shallower. Moving away from the forest edge, there is a transition zone where snow becomes progressively deeper. In contrast, the southern portion of the northwest field exhibits deep snow.

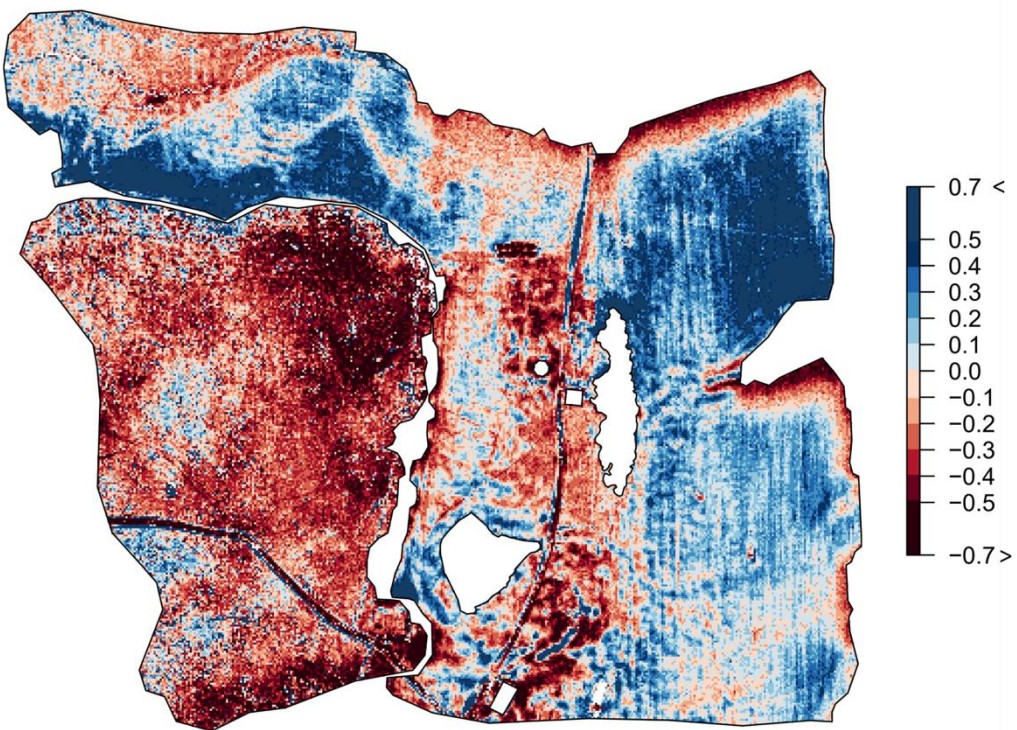

**Figure 6 The snow depth mean relative difference (MRD) map generated by averaging the eight relative difference maps from the UAS lidar-based snow depth maps from February 4th to March 7th.**

The relative difference snow depth maps for each date show that the spatial patterns of relative differences were fairly consistent throughout the study period (**Figure 7**). Generally, there was deeper snow in the northern part of the field (about 50% larger than the spatial mean), and shallower snow was found in forested areas as well as the central part of the field. Also, there was shallower snow along the northeastern boundaries of the field. These patterns were very clear during the

accumulation period before the peak snow depth around February 22nd. During the ablation period, consistent spatial patterns of the relative difference were observed despite the presence of patchy snow cover in some areas. These gaps emerged primarily due to differential melting, leading to sections with no remaining snow.

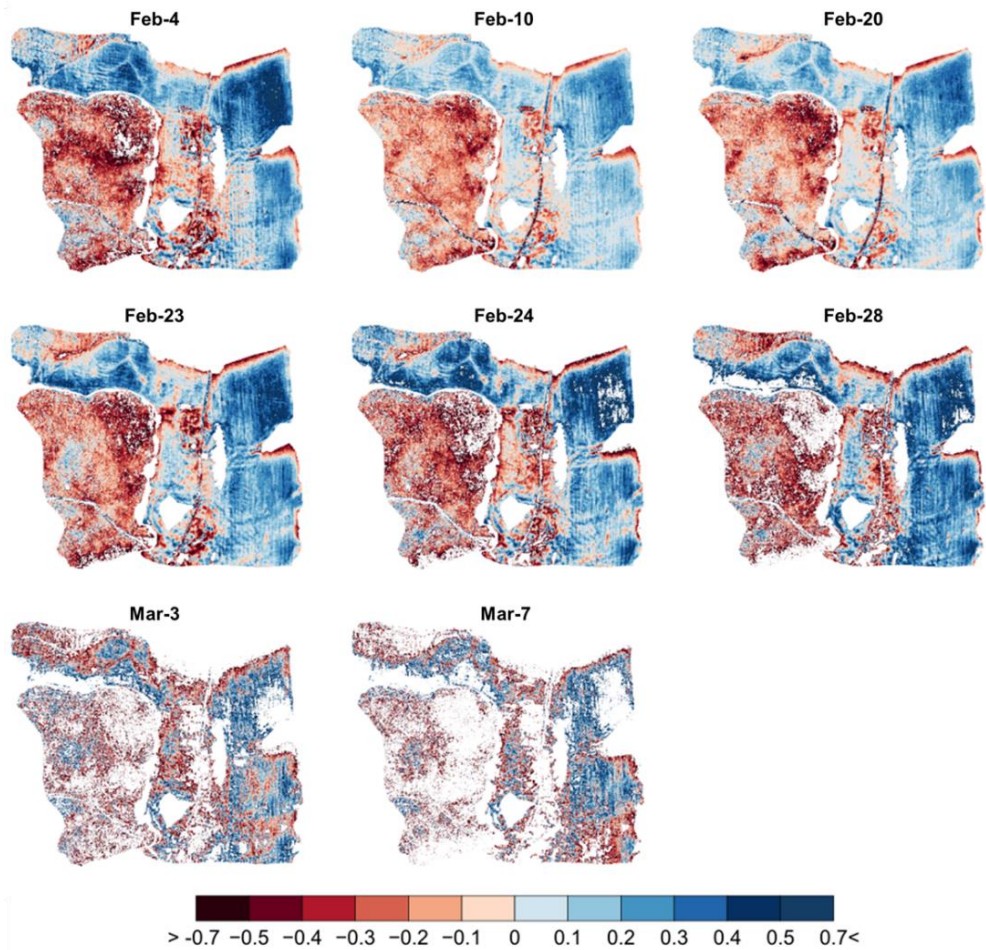

**Figure 7 Relative difference maps generated from the UAS lidar-based snow depth maps from February 4th to March 7th.**

**4.4 Physical Variables Characterizing Spatial Distribution of Snow Depth**

To evaluate the effect of physical land characteristics on the spatial distribution of snow depth, the MRD values were analysed with respect to five land and soil characteristic values (e.g., vegetation type, slope, shadow hours, $K_{sat}$, and SOM) over the study domain. Boxplots of MRD by physical feature are shown for the combined forest and field areas, field only, and forest only (**Figure 8**). Statistical significance results among groups, based on Kruskal-Wallis and Tukey tests summarized in **Tables S7** and **S8**. In the combined areas (i.e., forest + field), relative snow depth significantly differs by vegetation type. Coniferous forests have low MRDs (mean: -0.36) which indicates that snow in those areas is shallower relative to the spatial mean of snow

depth by around 36%. For the deciduous forest, the mean MRD is -0.2 with a wide interquartile range from –0.23 to 0.19. MRD values in the field are higher compared to the two forest types which ranged from –0.11 to 0.22 (mean: 0.08). For the combined areas as well as field and forest only, slope contributes to snowpack spatial patterns, even though the study area has a gentle slope (less than 20%). High MRDs are found in flat areas (0 – 5% slope) and gradually decrease with increasing slope. The effect of slope for the forest only area is relatively modest. The shadow hours show a clear but contradictory contribution to snow depth patterns in the field and forest only areas as compared to the combined area. When the field and forest are separated out, low MRDs are found in areas where shadow hours are short (e.g., less than 2 hours), and the MRDs gradually increase with increasing shadow hours. For the combined area, the highest shadow hours had the lowest snow depth, but this is likely the result of a mixed effect due to the dense shading in the coniferous forest. $K_{sat}$ shows little evidence of contributing to the spatial distribution of snow depth in the field, but there are distinct differences in MRDs among lower $K_{sat}$ groups in the forest. In the combined areas, MRDs tend to decrease with increasing the $K_{sat}$ values, except for the highest $K_{sat}$ group. Compared to $K_{sat}$, SOM exhibits a clearer pattern with MRD decreasing as SOM increases in both the combined areas and field analysis. The forest area does not display consistent MRD patterns with changes in SOM.

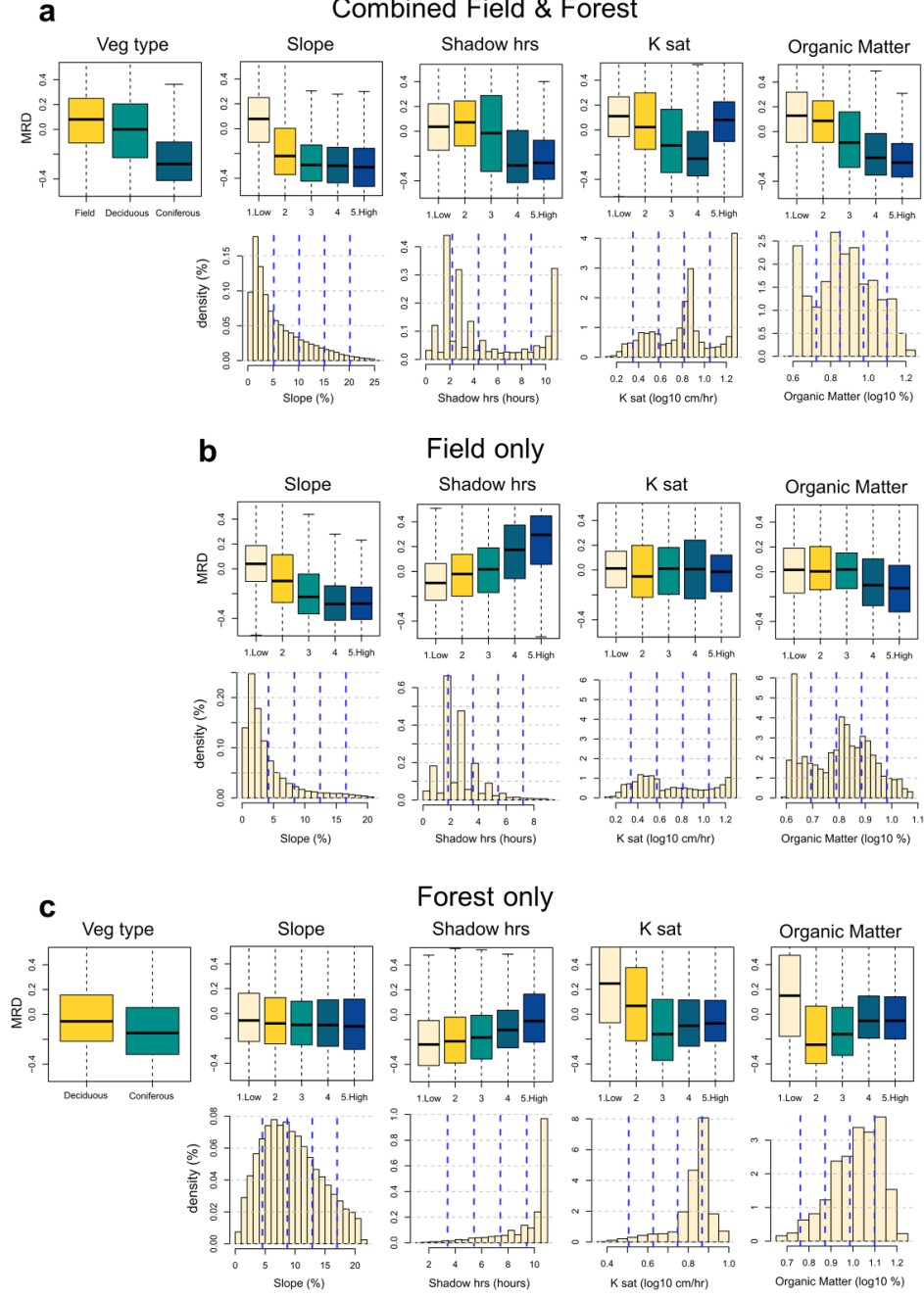

**Figure 8 Boxplots of the snow depth mean relative difference (MRD) by each physical feature (vegetation type, slope, shadow hours, Ksat, and soil organic matter) for (a) the combined areas (forest and field), (b) field, and (c) forest only. The 1-5 for each boxplot except for vegetation type represents the relative range of each physical variable in each area (For example, for slope in the combined areas, 1: 0-5 %, 2: 5-10%, 3: 10-15%, 4: 15-20%, and 5:20-25%).**

 **5 Discussion**

**5.1 Comparison with previous findings: UAS SfM and lidar snow depths**

The value of lidar datasets for capturing the horizontal and vertical structure of forests and snow cover depth is well established (Harder et al., 2020; Jacobs et al., 2021; Donager et al., 2021; Dharmadasa, Kinnard, and Baraër, 2022). However, the technology remains expensive, and data processing is complex. The lower cost of SfM techniques compared to lidar make them a valuable tool for conducting surveys of snowpack change over time (Fernades et al., 2018). Post-processing of RGB imagery is often less complex than lidar data processing, and a variety of SfM software is now available, including some open-source options. However, our study concurs with early findings that SfM accuracy for measuring forest snow depths still cannot match that of lidar (Donager et al. 2021).

In SfM processing, an insufficient number of valid tie points, used to stitch together overlapping images, may degrade the accuracy of SfM snow depth data (Harder et al., 2016). Due to its reliance on RGB optical imagery, overcast skies and poor lighting over relatively homogeneous snowpacks (e.g., fresh snow) make it difficult for SfM post processing software to identify a sufficient number of valid tie points (Bühler et al., 2016; Bühler et al., 2017; Harder et al., 2020; Revuelto et al., 2021b; Miller et al., 2022). A lower number of valid tie points and higher point uncertainty results in large data gaps and poor estimation of surface elevations. In this study, when there was relatively fresh snow and few features, there were gaps in the SfM snow maps in the eastern field on February 4th, 20th, and 24th. Areas in the field with a sufficient number of unique tie points showed better agreement between SfM and lidar-measured ground surface elevations than those with fewer tie points (**Figure S2**).

SfM performance is also lower in areas with dense vegetation where the ground surface is blocked by tree branches and canopies (Harder et al., 2020). Poor penetration of the forest canopy results in fewer overall ground returns and sparser point clouds. In Thompson Farm's mixed forest with dense underbrush, SfM post processing also experienced numerous tie point errors due to the lack of unique tie points. This is likely due to a plethora of thin tree branches and brush that appear as repetitive forest features. The features in these forests differ from forests with distinct trees and limited vegetation such as Harder et al.'s (2020) mixed lodgepole pine and subalpine fir forests. On all dates in the forest, the resampled 1 m x 1 m lidar data had fewer pixels with missing data than SfM (lidar = 0.4 – 1.6% NA, SfM = 3.7 – 70.5% NA). This error is apparent in the profile views of the ground surface returns for SfM and lidar from February 10th (**Figure S2**) where SfM-measured ground elevations had greater variability compared to lidar in the forest. Methods for reducing the errors associated with SfM in the forest are limited. Adjustments to survey techniques, such as changing the camera angle and flying at lower altitudes and speeds, and post processing workflows, such as making selection criteria less restrictive, may improve the number of points somewhat (Leendzioch et al., 2019). However, active remote sensing techniques, including lidar, have better penetration of the forest canopy than those which rely on passive sensing (Harder et al., 2020; Bühler et al., 2016).

The findings from our work are similar to previous studies which compared UAS SfM and snow probe measurements and found that the RMSE for snow depths is typically less than 31 cm in sparsely vegetated and alpine land cover types and increases to as much as 37 cm in areas with bushes, high grass, or forests (De Michele et al., 2016, Bühler et al., 2016, Avanzi et al., 2018, Belmonte et al., 2021). Studies using UAS SfM alongside rulers and snow stakes had an RMSE less than 14 cm in both forested and prairie land cover types (Fernandes et al., 2018; Harder et al., 2016). Findings were also similar for studies comparing UAS lidar to rulers or snow stakes which measured a RMSE less than 17 cm, with even lower RMSE values in shallow snow and in sunny areas (Harder et al., 2020; Feng et al., 2023; Koutantou et al., 2022). Lidar RMSE also tends to increase in vegetated areas regardless of the vegetation class or type (Harder et al., 2020). Much like Harder et al.'s (2016) observations of erroneously high SfM snow depth measurements several meters above the snow surface, we observed SfM measured snow depths greater than 150 cm in some forest locations. Conversely, our lidar measured snow depths never exceeded 25 cm, indicating a more consistent performance in forested areas. We also found that the SfM snow depths did not consistently agree with the lidar snow depths over the entire field and on all dates. On most dates, the difference in UAS-measured snow depths was close to 0 cm in the field. The best agreement occurred in the western portion of the field while the southeast and northwest portions had a larger amount of variability in measured values. The shadow hours and land cover type in the east and west fields are similar, however, the eastern field has a more gentle and less variable slope and fewer unique features (e.g., access road, USCRN station, pond, dirt piles, footprints) than the western field. The relatively homogenous features in the eastern field indicate that the difference between techniques is likely due to a lack of sufficient valid tie points for SfM. It is not clear what caused the differences between SfM and Lidar in the NW field that were not present in the other field areas. Unique features in the NW field are prevalent drainage patterns and shadowing that could be investigated further in the future.

While it is apparent that the accuracy of SfM-derived snow depth estimates cannot match that of lidar, the results of this study indicate that both techniques provide sufficient accuracy for monitoring the median change in shallow snow depths over time in flat, unforested land covers when there are a sufficient number of unique characteristics for SfM post processing. It is clear from the results of this study and previous ones that compared to in situ data, UAS lidar techniques produce lower errors and fewer data gaps than SfM, especially in forested land cover and over homogeneous snowpacks (Bühler et al., 2016; Bühler et al., 2017; Harder et al., 2020; Revuelto et al., 2021b; Miller et al., 2022). While UAS lidar may be the preferred technique in most cases, UAS SfM can still provide valuable information on changes in median snowpack depth across unforested areas at a relatively low cost and with less complex post-processing compared to UAS lidar. Regardless of the sampling technique used, the unique capability of UASs for measuring snowpack properties at the field scale and at a high temporal resolution makes them useful for observing snowpack evolution over time (Fernandes et al., 2018; Harder et al., 2020). Collection of in situ snow depth time series data is often time and cost prohibitive and may be especially challenging in complex or avalanche-prone terrain (Bühler et al., 2016; Harder et al., 2020). Monitoring snow

depth changes at the field scale provided insights into accumulation and ablation patterns across the entire study area, as well
as between different land cover types (e.g., forest and field).

By comparing maps of snow depth change with maps of physical variables at the site, specific factors influencing snowpack dynamics over the winter season were identified. Our findings highlighted that vegetation type is a dominant factor shaping snow depth patterns. In both combined and field-only areas, SOM showed a statistically significant relationship, with snow depth decreasing as SOM increased. Furthermore, shadow hours and slope were found to contribute to the spatial variability
of snowpack, even though the study area features relatively gentle slopes. These results demonstrate that high-resolution UAS observations are a powerful tool for quantifying snow pattern evolution. UAS observations are expected to advance the resolution of complex snow modeling by accurately capturing the physical relationships between snowpack dynamics and various climatic and topographic factors.

## 5.2 Physical variables at field scale

With a limited wind redistribution, time stability shows that the relative differences of the snowpack over the study region were generally stable throughout the accumulation and ablation periods. In addition to the previous findings that snowpack patterns are relatively consistent from year to year (Pflug and Lundquist, 2020; Revuelto et al., 2014), this study showed that fixed physical variables including vegetation, topography, and soil characteristics sufficiently control the spatial variations of snowpack throughout a winter period. The findings regarding the influence of vegetation and topographical factors on the
snowpack's spatial variability align with previous studies conducted (Currier and Lundquist, 2018; Deems et al., 2006; Trujillo et al., 2007). As compared to vegetation and terrain characteristics, few studies have examined the influence of soil characteristics on the snowpack. Our results indicate that snowpack depth decreases statistically significantly with increasing SOM (significant level < 0.01). This finding aligns with our previous study, which utilized maximum entropy modeling to analyze spatial variations of shallow snowpack over the same domain but during different periods (Cho et al., 2021). Even
though a clear relationship between $K_{sat}$ and snowpack was not found in this study (**Figure 8b**), it is acknowledged that soil thermal properties, such as the thermal conductivity of the soil underneath the snowpack, generally influence the rate of heat transfer between the snow and soil layers (Kane et al., 2001; Zhang, 2005). Also, the moisture content of the soil can affect the distribution of soil frost (Bay et al., 1952) and snowpack because the energy transfer at the snow-soil interface is controlled by wetness of the soil (Bay et al., 1952; Fu et al., 2018). Although spatial distribution of soil moisture is typically considered
to be constant (frozen) during winter, intermediate rainfall events and freeze-thaw cycles can dramatically change the spatial patterns of soil moisture and freeze-thaw states in regions having ephemeral snowpacks. This can be important because the thermal conductivity in frozen state is more sensitive to soil type than non-frozen condition, because the thermal conductivity of ice is four times larger than that of liquid phase (Penner, 1970). However, few studies have investigated how soil moisture patterns may control the spatial distribution of snowpack. This is likely because of the difficulty of measuring spatial
distributions of soil moisture and freeze-thaw states beneath the snowpack. Even though this study did not focus on it, future

investigations could measure spatial patterns of soil moisture and freeze-thaw states beneath the snowpack to quantify their interactions with the snowpack. A better understanding of the soil characteristics and their impact on the snowpack in various environments would help the snow community accurately predict and model snow distribution and snowmelt processes.

Although there are numerous studies that characterize temporal changes in the spatial distribution of snowpack across topographically uniform landscapes, encompassing both open field and forested environments (Hannula et al., 2016; Clark et al., 2011; Currier and Lundquist, 2018; Mazzotti et al., 2023), the concept of "time stability" (or "temporal stability") using relative difference values, as implemented in this study, has not been previously applied in snow hydrology. However, in the soil moisture community, numerous investigations that have examined temporal variability have been instrumental in developing robust validation sites and sampling strategies for satellite-based soil moisture assessments (Grayson and Western, 1998; Cosh et al., 2008; Brocca et al., 2009; Mohanty & Skagge, 2001; Jacobs et al., 2004). Similar to its utility in soil moisture studies, the integration of the time stability concept into snowpack analysis at the field scale could facilitate the identification of representative sampling locations and inform the design of sampling protocols for optimal spatial extrapolation. Extending this approach to diverse snow environments will contribute to quantifying spatio-temporal variability in snowpack, thereby enhancing the establishment of core validation sites for potential snow missions such as the Canadian Terrestrial Snow Mass Mission (TSMM; Derksen et al., 2019).

## 5.3 Limitations and future perspectives

Given that our investigation was conducted in a relatively uniform landscape characterized by a shallow snowpack, it is imperative to extend the analysis to encompass diverse plant functional types, climatic zones, and/or snow classes (Johnston et al., 2024; Sturm and Liston, 2021) to ascertain the generalizability of the findings. This is essential as snow depth distributions are influenced by terrain attributes and snow regimes (Clark et al., 2011; Currier and Lundquist, 2018). In contrast to the present study area, where spatial heterogeneity in snowpack is predominantly influenced by static terrain characteristics and vegetation cover, alpine and prairie regions experience variability due to wind-driven processes (Elder et al., 1991). Further investigation incorporating additional analyses of energy fluxes and meteorological parameters, including solar radiation, soil temperature and wind speed/direction, would enhance the comprehensiveness of the findings concerning the primary determinants of snowpack spatial variability across both static and dynamic variables.

Even though we analyzed spatial-temporal variability of the snowpack using well-validated UAS-based snow depth observations, this may not guarantee that the current findings also capture the SWE variations necessary for hydrologic applications. Snow density, needed to calculate SWE from snow depth, is affected by snow metamorphosis differently than snow depth. Snow density may change during snowmelt as water percolates into the snowpack and refreezes. Also, vegetation and soil characteristics strongly control turbulent and ground heat fluxes and impact snow properties including snow density (Pomeroy and Brun, 2001). Studies have used physics-based snow models such as SnowModel (Liston and Elder, 2006), Crocus (Vionnet et al., 2012), and Flexible Snow Model (FSM2; Essery et al., 2024) to understand those physical processes that cause snowpack spatial variations. However, observational approaches focusing on spatial distribution of SWE are quite

limited because only now are sensing techniques emerging that directly observe the spatial distribution of snow density with a UAS system (McGrath et al., 2022). A potential future direction is to develop reliable, high-resolution SWE maps by integrating emerging techniques such as lidar and gamma-ray spectrometry (Harder et al., 2024), enabling the quantification of SWE spatial distribution across diverse snow environments. Another direction could involve employing an integrative approach using physical models to maximize in situ and UAS snow observations through data assimilation and/or novel interpolation methods, utilizing machine (or deep) learning approaches.

## 6 Conclusion

In this study, UAS lidar and SfM snow depth measurements were assessed using ground-based magnaprobe data and then used to confirm that spatial patterns of snowpack depth are temporally stable. Lidar demonstrated superior performance compared to SfM when evaluated against in situ observations, exhibiting lower errors. Both UAS techniques exhibited lower errors in field settings (lidar MAD = 3.5 cm, SfM MAD = 4.0 cm) than in forested environments (lidar MAD = 6.3 cm, SfM MAD = 31.4 cm). As expected, differences between lidar and SfM snow depths were more pronounced in forested regions (MAD = 55.7 cm), with SfM often registering anomalously deep snow depth values. The spatial distribution of snow depth captured by lidar remained consistent throughout the study period. For the entire study area, deeper snow was found in the field, in locations having shallow slopes and lower soil organic matter. Within the field, snow deepened with increasing shadow hours. When examining combined landscapes including forests and fields, we observed that the spatial distribution of snow depth was predominantly shaped by the type of vegetation present. Within the field, the spatial distribution of snow depth tracked with relatively modest local slope variations and shadowing effects at the forest-field edge. As ephemeral snow conditions expand in a warming climate, these results are valuable for effectively comparing UAS and in situ sampling techniques for ephemeral, shallow seasonal snowpacks. It is also expected that this study contributes to the enhancement of land surface and snow models by offering insights into parameterizing sub-grid scale snow depths, downscaling coarse-scale remotely sensed snow observations, and comprehending snowpack evolution at the field scale, particularly in ephemeral snow environments.

*Data availability*. The UAS lidar and SfM photogrammetry snow depth maps, along with topographic variables developed in this study, will be available for download from the Hydroshare repository once this manuscript is accepted and published (currently being set up with an ODC Attribution license for unrestricted access). Daily precipitation and mean temperature data are available from the NOAA U.S. Climate Reference Network at https://www.ncei.noaa.gov/access/crn/sensors.htm?stationId=1040.

*Author contributions*. EC, MV, JMJ, and AH conceptualized the research, conducted the formal analysis, and wrote the initial draft. FS, MP, and CW assisted with fieldwork and investigation, providing technical and scientific inputs. JMJ supervised the project. All authors reviewed and edited the paper.

*Competing interests*. On behalf of all authors, the corresponding author states that there is no conflict of interest

*Acknowledgments*. This study is based upon work supported by the U.S. Department of Defense Engineer Research and
Development Center's Broad Agency Announcement (BAA) #W912HZ20BAA01 and the U.S. Army ERDC's Cold Region
Research and Engineering Laboratory (CRREL) under contract number W913E5-21-C-0006. Any opinions, findings, and
conclusions or recommendations in this material are those of the authors and do not necessarily reflect the views of the Broad
Agency Announcement Program and the ERDC-CRREL. DISTRIBUTION A: Approved for Public Release. Distribution is
Unlimited. The authors are grateful to Elizabeth Burakowski, Mahsa Moradi Khaneghahi, and Holly Proulx for helping data
collection and providing valuable comments.

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
