# Peer review of "Characterizing Spatial Distribution of Field-Scale Snowpack using Unpiloted Aerial System (UAS) Lidar and SfM Photogrammetry"

_EGUsphere, 2024_

## Referee Comment (RC2)

**Manuscript Title:** Characterizing Spatial Structures of Field-Scale Snowpack using Unpiloted Aerial System (UAS) Lidar and SfM Photogrammetry

**General review:** Cho and other authors measure the spatial heterogeneity of snow across a study plot in New Hampshire while evaluating the performance of UAS structure for motion (SfM) photogrammetry against lidar and in-situ observations during one snow season. This effort was conducted in New Hampshire, USA, and the study plot includes both a forested region and open field. The authors determined that the open areas tended to have deeper snow than forested areas and that lidar and SfM generally performed better, compared to in-situ observations and each other, in the open areas of the study plot. Static landscape variables, such as vegetation type and slope, impacted the distribution of snow consistently across the snow season. This project provides a new evaluation of UAS SfM as a tool to measure snow depth (a lower cost option compared to lidar). Much of the analysis, and thus manuscript text, includes a very clear description of the methodology used. However, the presentation of results in figures could be refined for clarity, and the results also require a more in-depth discussion around the performance of the select UAS tools, given the first two objectives of the study. The following line-by-line comments, which vary in "major" versus "minor" feedback, should provide more clarity and direction regarding this review, with the hope of better emphasizing the importance and value of this work.

**Line-by-line comments**
Abstract: Throughout, it is initially unclear and confusing what "spatial structure" is referring to. The words "patterns" and "spatial variability" are only used at the beginning of the introduction, which provide more clarity. Suggest briefly including a definition in the abstract, given the use of "spatial structures" in the manuscript title. Otherwise, suggest replacing "structure" with "variability, ""heterogeneity," or "distribution," which are more commonly used in the literature (including in the citations provided within this manuscript), whereas "structure" is often associated with the vertical microstructure of the snowpack.

Line 27-28: It would be beneficial and more complete to report, at the very least, the direction of the correlations.

Line 36: Here "snowpack structure" is ambiguous, where, to some readership, the term insinuates the vertical, microscale structure of the snowpack.

Line 70: Unclear what "these transition periods" are referring to. Please define.

Line 73-75: It would be impactful for the authors to include why this type of forest/snowpack was chosen. For example, a number of the previously cited UAS works take place in other climates/forest types.

Line 89: Reference error. There are a number of these throughout – also associated with figure references – thus I will only note this one.

Section 2: Suggest further emphasis on why this area might be ideal for this type of study (shallow snow depths, type of forest, historical data, etc.). Obviously, there are many other locations which offer open versus forested regions.

Figure 1: It would potentially eliminate preemptive readership questions if the authors stated that the derivation of the variables shown in Figure 1b-g is explained in the following section (3).

Table 1: The concept of a mixed forest is not also shown in Figure 1b. Is this referring to a blend of both coniferous and deciduous trees? If so, it is unclear the fraction of coniferous and deciduous used to determine the mixed forest area (50/50?).

Line 131: Can the authors provide insight on which ground conditions resulted in more returns versus less?

Line 174: Suggest a Sturm citation for the magnaprobe.

Line 180: As written, it is unclear if the 9 in-situ measurements were at random within the 1x1m or consistent across each survey. And can the authors please elaborate on why full sampling was not conducted during each flight (time/personnel constraints)?

Figure 2: It is currently challenging to determine the main takeaway of this figure – is it simply to observe the timeseries or to compare across the field versus forested areas? For example, it appears that air temperature and precipitation/cumulative precipitation are the same, which would make sense given data availability, but is thus redundant. It is particularly challenging to follow the 3x y-axis labels on the right side of each figure. Suggest reformatting as a sequence of timeseries – with only 1x precipitation/cumulative precipitation panel, 1x air temperature panel, and potentially 2x snow depth panels for each area (forest versus field), including the in-situ observations.

Figure 3: Are N-values the same across the two panels? Please add.

Line 278-279: Suggest including the sub-areas of the field when introducing the field and an explanation as to why there the authors created a division here (e.g., what led to the decision making for a NW vs. W vs. E sub-area of the field?).

Figure 5: Suggest a more intuitive color scheme for snow depth. The difference color scheme makes sense (negative = red vs. positive = blue). For just snow depth, suggest purple leading to blue and then red (or something similar where red is not a color in the middle of the color bar).

Figure 8: It is unclear what 1-5 (low to high) represents – this should be stated in the figure caption. Further, are there any statistical differences? If so, please note here and in the paragraph above with relevant p-values.

Line 338: Here and throughout the manuscript, the terms "modestly" and "higher" read subjectively and would be more impactful if numerical values accompanied them and/or if there was a defined threshold for what the authors considered "modest" versus "high."

Discussion: From the results section (e.g., Figure 4a, Figure 5), readers are led to believe that using SfM for snow depth generally is not a feasible option (without significant uncertainty) except for the west side of the field, and there isn't much of an explanation as to why. It is unclear what makes the west side of the field different from the rest of the field? Differences to the forested portion of the study plot are perhaps more obvious but are also not stated explicitly. Further, what might the authors suggest doing differently to reduce the numerous erroneous SfM measurements? The only plausible explanation currently provided is insufficient number of point clouds. It is stated in the introduction that this methodology is still an emerging one, thus this seems like an opportunity provide insight into how UAS SfM for snow depth measurements may still evolve.

Line 344: It would be helpful to connect the western portion of the field in this study to the subsequent sentences on past studies – e.g., does the western portion have a different vegetation type or other static landscape characteristic/combination of note (nothing particularly stood out in Figure 1)?

Line 353: Can the authors indicate what likely caused the erroneous values of 150+ cm of snow depth as measured by SfM?

Conclusion: Suggest restating the error values when referring to "lower error"

Line 431-433: Suggest explicitly restating the relationship – e.g., x vegetation type leads to deeper [or shallower] snow depth.

---

## Referee Comment (RC3)

**General Comments**

The authors present a generally well-written and well-referenced study comparing two UAS-based snow remote sensing techniques (Lidar and Structure from Motion) across different landscapes in New Hampshire, USA. I appreciate that the study was conducted over relatively shallow snowpacks (~tens of centimeters), which I feel are underrepresented in the literature and pose unique challenges for obtaining accurate snow depth measurements. I also like that the authors incorporated the Relative Difference concept to analyze this timeseries of spatially distributed snow depth measurements. I was not familiar with this concept but I think it led to an interesting framing of the results.

While the introduction and methods sections are generally clear and easy to follow, there is room for improvement in the presentation of results and subsequent discussion. As noted in the Specific Comments below, I feel that the discussion is lacking critical engagement with some of the more complicated findings from this study. In particular, the SfM results do not inspire much confidence for the technique overall. I appreciate that the authors did not try to hide these larger errors, but there is little discussion around the potential sources of those errors, or suggestions for how these errors might be avoided in future studies (if that is even possible). This study could be more impactful to the broader snow/UAS community if some of these topics were explored more deeply in the discussion.

**Specific Comments**

Line 21: I am a bit surprised to see a range of errors for SfM but not lidar here, since they both have ranges on the next line. Perhaps an accidental omission?

Line 42-49: This paragraph would benefit from some additional discussion/more precise language related to spatial variability (Line 42), spatial patterns (Line 43), and hydrologic patterns (Line 44). With multiple phrases present it is unclear if these are different concepts, and which properties of the snow are relevant. E.g. if snow stratigraphy is variable over some length scale but snow depth is not, is that spatially variable snow but not a hydrologic pattern? Please clarify here and implement similar changes throughout the paper, e.g. Sections 4.3, 4.4, 5.

Line 55: Please quantify (at least approximately) "small-scale snow patterns" – tens of meters? It would also be helpful if you could relate this phrase to "field or local-scale snow features" at the beginning of this paragraph (Line 50).

Line 69-70: "e.g., forest and fields" – I think more than just these landscapes. UAS technology has also enabled rapid progress in complex/mountainous terrain for example. I suggest removing this parenthetical statement from the end of the sentence.

Line 70: "these transition periods" – unclear, previous sentence mentions "the entire snow period"

Line 75: Please clarify why you only investigate snow depths less than 35 cm. Is this just based on the datasets you collected or is there a particular motivation for snow depths below this threshold?

Figure 1: Consider widening the color bars in panels c-g. Panel b is wider and easier to discern the colors.

Table 1 caption: The information about the sampling strategy in each 1x1 m grid cell seems better suited for the main text.

Line 149-152: With the in situ sampling strategy there are always 9 Magnaprobe measurements for the average snow depth. Is it possible to provide an approximate range of the number of lidar ground returns within each 1x1 m square? How does this number compare to the 9 Magnaprobe measurements, and is it relatively consistent throughout the study or does it change with time, landscape, etc?

Line 170: "following the same procedure as the lidar" – much work went into processing the lidar data. If you mean that you subtracted the snow-on and snow-off SfM maps to get snow depths, I suggest writing that explicitly here.

Line 180: Please clarify if the 9 measurements were taken in the same pattern at every grid cell ( I am envisioning a 3x3 pattern hitting all 4 corners and the middle of the cell, but this is worth specifying).

Line 183: does the less accurate GPS (~centimeter scale) matter compared to the RTK-driven image geotagging (listed as sub-centimeter in Line 155)?

Line 210-211: Did you factor in a potential change in the shadow hours between February 4 and March 7 based on changing solar angles? Maybe this is negligible for the results of this study but would be good to clarify either way.

Line 230-249: Please clarify in this paragraph if USCRN precipitation values refer to snow depth or snow water equivalent. It would be helpful to specify in the first mention of the station (Lines 190-192) what sensor is available to measure precipitation and how you convert that to snow depth (assumed density?) if there is no dedicated snow depth sensor on the station.

Line 242-246: This is probably fine since we expect a fair amount of variation across the transects with a fairly shallow snowpack overall. But how did the field camera measurement compare with the closest 1-2 magnaprobe grid cells? I think that would

provide more meaningful information than a comparison with the average across the entire transects. Similar comment for the forest site (Line 251-253).

Figure 2: I'm having a difficult time understanding what's going on here with 4 shared y-axes. Some points of confusion:

- Cumulative precipitation doesn't start at 0, which implies it's cumulative from some date earlier than the start of the field campaigns. Maybe the start of the water year? But the field cameras at both sites imply 0 snow depth at the beginning of this timeseries. To me the cumulative precipitation does not provide any meaningful information in the context of these field campaigns. It also looks like the same curve in both subplots so I suggest at least removing one of the redundant curves, if not both.
- I assume precipitation (mm) is in reference to snow water equivalent when the air temperatures are below 0 C. I suggest stating this explicitly somewhere. Is there a snow depth sensor on the USCRN station?
- Caption states "UAS-based measurements represent average of all samples" but it also looks like some error bars are included, which are often covered up by different colored error bars from a different location. I suggest either removing the error bars completely and just stick to an average, or find some way to stagger the different sampling locations to prevent overlap. Also specify if the error bars represent IQR, +/- 1 standard deviation, etc.

One way to make this information clearer (with fewer shared axes) might be to have one subplot for temperature and precip (since the data are the same at both sites) and then separate subplots for the field and forest snow depth measurements.

Line 258-259: "All snow observing methods were able to distinguish that the average snow depth was slightly deeper in the forest than the field." Is this a mixup of forest and field? Compare reported snow depths in Line 236-238 as well as results in Section 4.3.

Figure 3: Please note in the caption the different axis limits between the subpanels.

Figure 4: I like the subpanel showing the color coding of the different fields, but perhaps remove the forest outline as it took me a minute to realize those data are not included on the left figures. Also it is difficult to discern the difference between the solid Lidar lines and the dashed SfM lines in 4c.

Figure 5: Personally I am not a big fan of the snow depth color bar and I'm not sure that it will be colorblind friendly. Did you try using a simple white -> blue gradient for snow depth? Your choice in the end, this is just a suggestion. The red -> blue gradient makes sense for the difference maps.

Figure 6/7: I suggest switching the order of these figures. The MRD map is a slightly easier concept for me to grasp and leads nicely into the individual RD maps. Plus it's nice to see the larger, detailed map before the smaller subpanels in the current Figure 6.

Figure 8: I really like this layout. Keeping five consistent boxplots is helpful across the different variables, and I appreciate the distributions below showing how they divide into the different boxplots. However, the discussion in Sections 4.4 and 5.2 would be strengthened if you could bring in some measure of statistical significance, e.g. Line 321-323 "In the combined areas, the MRDs seem to decrease with increasing the Ksat values, except for the highest Ksat group, there are no significant patterns of MRDs when field areas are analyzed only." – how can you be certain there are no significant patterns without a statistical test? Perhaps look into notched boxplots as a starting place, but there are other possibilities here.

Section 5.1: To me this discussion is lacking critical engagement with some of the more complicated findings from this study. I'm not sure I agree that "It is clear from the results of this study and previous ones that both UAS SfM and lidar techniques provide a viable method for monitoring snow depth change across many land cover types." (line 358-359) based on the SfM results in Figures 3 and 4 where the SfM depths are anywhere from 2-10 times larger than the in situ measurements. I doubt there are many applications where errors of this magnitude are acceptable. Additionally it doesn't seem feasible to rely on the SfM technique in forested areas based on all the missing data in Figure 4. Can you expand upon either of these? You briefly mention overcast skies possibly affecting SfM data collection (line 337) but this doesn't seem to explain why the SfM snow depths in the western field had much better agreement than the E and NW fields (Figure 4). What was the vegetation like in the fields? Was it fully buried by snow or partially extending above the snowpack? Could there be GPS/processing errors affecting the final results? Including individual photos from the SfM photosets could help illustrate some of the challenges.

Section 5.2: Similar to a comment above, this section would be stronger if the relationships between physical variables and snow depth could be quantified statistically.

Lines 409-422: In the description of the in situ data collection you noted that one SWE sample was collected in each grid cell. Did you try any analysis with those measurements?

**Technical Corrections**

Line 26: all areas → both areas?

Line 88-89: Missing reference

Line 122: acronym IR not defined

Line 154: acronym CMOS not defined

Line 182: remove superscript formatting from "antenna"

Line 185-186: Typo in personal communication date? Data for this study collected in 2021 but personal communication listed as 2023

Line 200: Missing reference

Line 200-201: "snow-off"

Line 231: Missing figure number

Line 242: Missing figure number

Line 277: Missing reference

Line 283: Missing reference

---

## Author Response (AR1)

**Reviewer 1**

This study introduces a novel dataset of snow depth (HS) observations for a small study site with mixed vegetation and open, non-vegetated areas. HS was measured using both, LiDAR and SfM systems mounted on a UAV at different dates during one snow season. The study further evaluates the HS maps against different manual in-situ reference measurements and compares the LiDAR and SfM products against each other. The dataset is finally analysed using the relative distance concept and by comparing the mean relative distance (MRD) to 5 different spatial features describing soil and terrain characteristics.

Novel datasets of HS distribution using UAV-based remote sensing are strongly encouraged and of great importance for the hydrologic community (especially if the presented data sets will be made publicly available). Only a few other data sets exist that provide repeated, high-resolution HS observations from UAVs. For cold-maritime environments in in the north-eastern US, no comparable data sets exist (Except for the data set presented for the same study site by Jennifer M. Jacobs et al. (2020). The data therefore has the potential to provide new insights into the specific drivers for the spatial distribution of HS in such environments and is scientifically significant. The performed analysis, however, lacks a clear scientific aim and does not give new insights. The findings of the manuscript, that i) SfM provides less reliable observations with larger data gaps underneath the forest canopy and ii) HS distribution being influenced by topography and vegetation are not new to the scientific community. See for instance Harder et al. (2020) that performed a comprehensive comparison of SfM vs LiDAR HS maps derived from UAVs and Mazzotti et al. (2023) that discussed the different drivers of HS variability at high spatial resolutions. The presented method moreover lacks a co-registration of the snow-off and snow-on maps. I see the greatest potential of your manuscript and data to further discuss the relative difference maps and analyse their temporal stability quantitatively. Understanding the consistency of spatial HS patterns is an important aspect of current snow research (Geissler et al., 2023; Pflug & Lundquist, 2020).

*Thank you for your constructive feedback and the specific suggestions on our manuscript. We have addressed each of your comments and carefully revised the manuscript accordingly. The referenced literature was instrumental in enhancing the positioning of our paper within the community.*

**Major Comments:**

Overall Style:

1. The manuscript contains erroneous links to figures and references.

2. The manuscript overall has a good structure but could benefit from a careful proofreading (See also minor comments) and a more scientific writing style.

3. Some figures are not satisfactory and need to be improved (See minor comments).

*Thank you for your feedback on the manuscript's overall style, references, and figure quality. We appreciate your detailed observations, and we have carefully revised the manuscript in response to each point.*

Method:

1. I encourage the authors to perform a co-registration of the snow-off and snow-on surveys. If there are no snow-free areas within the snow-on surveys to perform a co-registration with, maybe a coregistration of the point clouds (or at least the elevation models) for each survey using SfM and LiDAR could help to reduce systematic over-/underestimation of the products. Check https://github.com/jgenvironment/cluster_snow for ideas on how to perform a three-dimensional co-registration of point clouds using vegetation. Did you check whether your snow-off surveys are comparable between both systems?

Thank you for this comment. All of the surveys were co-registered using ground control points surveyed with a Trimble Geo7X GNSS Positioning Unit and Zephyr antenna. Linear, horizontal, and vertical shifts were applied to align all digital elevation models to the GCPs. The authors have added the following text in Section 3.1 to clarify this:

"The GCPs were surveyed in using a Trimble© Geo7X GNSS Positioning Unit and Zephyr™ antenna with sub-centimeter accuracy."

"GCPs surveyed using the base/rover equipment were used to co-register the UAS data. Linear, horizontal, and vertical shifts were applied to align all SfM and lidar DEMs to the GCPs."

2. Subsequently, I would be interested in a transect – plot (of the underlying point clouds) that could give a better understanding of i) where the differences between SfM and LiDAR originate and ii) give an idea of the sub-canopy point density of the two products. Providing the point densities of the different products for the different surveys is essential. (Overall, forest, open).

The authors created a transect plot (Figure S2) and summary tables (Tables S3-S6) of the point counts for SfM and lidar similar to Harder et al. (2020) to address this comment. This content has been added to the supplement.

[Figure]

**Figure S2** Comparison of ground surface returns for SfM and lidar surveys from February 10, 2021. Panels on the right-side show profile views of the ground returns from lidar (blue points) and SfM (red points) point clouds. The green transect (a – b) shows the greater variability in SfM ground elevations compared to the lidar in the forest. The blue transect (c – d) shows that SfM and lidar agree well in the field when the SfM software identifies an adequate number of tie points. The purple transect (e – f) shows how fewer SfM tie points in the field results in poor dense cloud reconstruction in those areas.

**Table S3** Lidar ground return point count (number of ground returns per 1 m x 1 m pixel) statistics by land cover type. The flight on 4/2/2021 was the snow-off baseline flight.

| Location | Date | Mean | Median | Standard Deviation | 75th Percentile | 25th Percentile |
|---|---|---|---|---|---|---|
| Field | 2/10/2021 | 1231.7 | 1272 | 197.7 | 1373 | 1116 |
| Field | 2/20/2021 | 1062.8 | 1080 | 170.2 | 1179 | 965 |
| Field | 2/23/2021 | 900.6 | 911 | 164.3 | 1006 | 812 |
| Field | 2/24/2021 | 851.2 | 873 | 173.1 | 962 | 770 |
| Field | 2/28/2021 | 893.1 | 930 | 207.9 | 1021 | 817 |
| Field | 2/4/2021 | 1059.2 | 1080 | 183.9 | 1187 | 951 |
| Field | 3/3/2021 | 527.1 | 566 | 207.1 | 668 | 430 |
| Field | 3/7/2021 | 748.5 | 801 | 258.3 | 930 | 620 |
| Field | 4/2/2021 | 664.1 | 669 | 218.6 | 803 | 537 |
| Forest | 2/10/2021 | 442.2 | 440 | 186.9 | 580 | 299 |
| Forest | 2/20/2021 | 299.7 | 289 | 151.0 | 402 | 183 |
| Forest | 2/23/2021 | 215.2 | 199 | 130.0 | 299 | 114 |
| Forest | 2/24/2021 | 179.8 | 164 | 116.4 | 255 | 87 |
| Forest | 2/28/2021 | 145.7 | 127 | 105.1 | 212 | 61 |
| Forest | 2/4/2021 | 264.0 | 252 | 139.4 | 357 | 156 |
| Forest | 3/3/2021 | 125.1 | 111 | 83.8 | 177 | 59 |
| Forest | 3/7/2021 | 187.0 | 174 | 108.1 | 257 | 103 |
| Forest | 4/2/2021 | 178.3 | 162 | 112.0 | 251 | 89 |

**Table S4** SfM ground return point count (number of ground returns per 1 m x 1 m pixel) statistics from derived point clouds by land cover type. Accuracy of SfM point clouds depends on the RTK geotagging as well as the software's ability to reconstruct the 3D model from tie points. The flight on 4/2/2021 was the snow-off baseline flight.

| Location | Date | Mean | Median | Standard Deviation | 75th Percentile | 25th Percentile |
|---|---|---|---|---|---|---|
| Field | 2/4/2021 | 390.6 | 331 | 249.6 | 461 | 325 |
| Field | 2/10/2021 | 322.4 | 318 | 39.5 | 319 | 317 |
| Field | 2/20/2021 | 238.2 | 308 | 116.4 | 311 | 183 |
| Field | 2/23/2021 | 322.5 | 312 | 69.3 | 315 | 310 |
| Field | 2/24/2021 | 309.2 | 307 | 21.2 | 308 | 306 |
| Field | 2/28/2021 | 319.8 | 315 | 51 | 317 | 314 |
| Field | 3/3/2021 | 325.5 | 317 | 48.5 | 319 | 316 |
| Field | 3/7/2021 | 317 | 315 | 37.2 | 316 | 314 |
| Field | 4/2/2021 | 327.6 | 323 | 46.4 | 324 | 321 |
| Forest | 2/4/2021 | 369.6 | 132 | 487 | 640 | 0 |
| Forest | 2/10/2021 | 525.8 | 453 | 265 | 656 | 341 |
| Forest | 2/20/2021 | 27.5 | 0 | 75.5 | 1 | 0 |
| Forest | 2/23/2021 | 692.6 | 639.5 | 459.6 | 907 | 406 |
| Forest | 2/24/2021 | 148.3 | 135 | 124.5 | 281 | 16 |
| Forest | 2/28/2021 | 1053.6 | 909 | 798 | 1471 | 438 |
| Forest | 3/3/2021 | 616.6 | 492 | 372.3 | 775 | 339 |
| Forest | 3/7/2021 | 743 | 569 | 496.7 | 965 | 360 |
| Forest | 4/2/2021 | 477.7 | 415 | 260.6 | 601 | 328 |

**Table S5** Lidar ground return point count (number of ground returns per 1 m x 1 m pixel) statistics by forested land cover type. The flight on 4/2/2021 was the snow-off baseline flight.

| Location | Date | Mean | Median | Standard Deviation | 75th Percentile | 25th Percentile |
|---|---|---|---|---|---|---|
| Deciduous | 2/10/2021 | 485.7 | 487 | 173.8 | 612 | 356 |
| Deciduous | 2/20/2021 | 333.4 | 325.5 | 143.3 | 430 | 228 |
| Deciduous | 2/23/2021 | 240.9 | 228 | 127.8 | 323 | 145 |
| Deciduous | 2/24/2021 | 203.3 | 192 | 114.8 | 278 | 117 |
| Deciduous | 2/28/2021 | 167.3 | 154 | 104.7 | 233 | 86 |
| Deciduous | 2/4/2021 | 292.9 | 284 | 134.6 | 381 | 192 |
| Deciduous | 3/3/2021 | 138.8 | 128 | 83.8 | 192 | 74 |
| Deciduous | 3/7/2021 | 206.3 | 195 | 106.4 | 276 | 125 |
| Deciduous | 4/2/2021 | 198.8 | 187 | 111 | 272 | 113 |
| Coniferous | 2/10/2021 | 291 | 273 | 147.7 | 380 | 182 |
| Coniferous | 2/20/2021 | 182.4 | 162 | 114.1 | 250 | 96 |
| Coniferous | 2/23/2021 | 126 | 106 | 92.8 | 174 | 57 |
| Coniferous | 2/24/2021 | 98.2 | 79 | 78.9 | 136 | 40 |
| Coniferous | 2/28/2021 | 71 | 54 | 64.5 | 99 | 24 |
| Coniferous | 2/4/2021 | 163.5 | 143 | 104.6 | 223 | 86 |
| Coniferous | 3/3/2021 | 77.3 | 61 | 63.8 | 108 | 30 |
| Coniferous | 3/7/2021 | 120.3 | 101 | 84.8 | 165 | 57 |
| Coniferous | 4/2/2021 | 107.4 | 88 | 82.4 | 149 | 46 |

**Table S6** SfM ground return point count (number of ground returns per 1 m x 1 m pixel) statistics from derived point clouds by forested land cover type. Accuracy of SfM point clouds depends on the RTK geotagging as well as the software's ability to reconstruct the 3D model from tie points. The flight on 4/2/2021 was the snow-off baseline flight.

| Location | Date | Mean | Median | Standard Deviation | 75th Percentile | 25th Percentile |
|---|---|---|---|---|---|---|
| Deciduous | 2/10/2021 | 512.9 | 434 | 252.9 | 637 | 336 |
| Deciduous | 2/20/2021 | 32 | 0 | 80.9 | 4 | 0 |
| Deciduous | 2/23/2021 | 666.4 | 619 | 437 | 878 | 392 |
| Deciduous | 2/24/2021 | 164.8 | 172 | 123.7 | 294 | 33 |
| Deciduous | 2/28/2021 | 1035.6 | 892 | 788.2 | 1456 | 428 |
| Deciduous | 2/4/2021 | 378.5 | 159 | 481.9 | 658 | 0 |
| Deciduous | 3/3/2021 | 624 | 504 | 369.5 | 790.75 | 342 |
| Deciduous | 3/7/2021 | 732.7 | 549 | 497.9 | 950 | 351 |
| Deciduous | 4/2/2021 | 480.9 | 413 | 252.3 | 602 | 328 |
| Coniferous | 2/10/2021 | 570.6 | 521 | 299 | 718 | 370 |
| Coniferous | 2/20/2021 | 12 | 0 | 49.4 | 0 | 0 |
| Coniferous | 2/23/2021 | 782.6 | 709 | 519.9 | 1001 | 463 |
| Coniferous | 2/24/2021 | 90.6 | 35 | 108.9 | 166 | 0 |
| Coniferous | 2/28/2021 | 1116.1 | 967 | 828.4 | 1522.75 | 486 |
| Coniferous | 2/4/2021 | 339 | 72 | 503 | 559 | 0 |
| Coniferous | 3/3/2021 | 591 | 457 | 380.9 | 720.75 | 329 |
| Coniferous | 3/7/2021 | 779.1 | 635 | 490.9 | 1000 | 400.5 |
| Coniferous | 4/2/2021 | 466.6 | 425 | 287.4 | 600 | 327 |

3. I suggest removing the different validation strategies (cameras vs. magnaprobe) from your manuscript, as the number of camera locations is not sufficient to give reliable results. Moreover, as you show, the differences are negligible. Your magnaprobe measurements are important to provide the overall accuracies of your data sets but are too small in number to provide reliable insights into potential deficits of your HS maps.

Thank you for this suggestion. Based on your feedback, the authors have decided to remove the comparison between the in-situ measurement types and instead include a brief summary of the UAS SfM and lidar validation based on the magnaprobe data alone. The study objectives presented in the abstract and introduction have been updated and additional text was added to section 4.1 to explain that the cameras were not used for validation due to their limited spatial extent compared to the magnaprobe sample locations. Additional text was also added to section 4.1 to discuss the potential sampling bias for the magnaprobe data (i.e., overprobing) in Figure 3b. All statistics presented in section 4.1 and Supplementary Table S2 (formerly Table S1) now only reference the magnaprobe data. The field camera data is retained in the updated version of Figure 2 as it provides the only continuous measurement of snow depth throughout the field season.

4. Instead, you could further examine the relative difference maps, discuss their temporal stability quantitatively and relate areas of persistent relative differences with areas of varying relative differences to your topographic and soil features. This is where I see the greatest potential of your work.

We agree. We have reduced the text in the sections on validation and are expanding the section that describes the relative difference maps to include content on the three field areas.

5. Secondly, I would try to work out explicitly where and when SfM is a suited method to measure HS and what strengths and weaknesses of this sensor are. Did you find some features/flight conditions that impacted the SfM more than LiDAR?

Thank you for this suggestion. We have revised section 5.1 to include more discussion on the relative performance of the two techniques and the limitations of SfM for the study region. We added more information on the greater potential for SfM tie point errors in forested/vegetated areas, over homogenous snowpacks (e.g., fresh snow), and in sub-optimal lighting conditions. Citations are presented for each of these conditions. In addition, further discussion was added to describe problem areas in the field at the Thompson Farm site (e.g., northwest field, east field) and potential causes for the larger differences in these areas (e.g., lack of unique features, fresh snow).

Introduction and Discussion:

1. As noted above, I would focus your work i) on the comparison of SfM and LiDAR and ii) the subsequent analysis of relative differences, including the quantitative assessment of temporal stability. For i), I would explicitly state for which conditions you can recommend using SfM. Obviously, your results show that SfM is not capable of measuring sub-canopy snow ($r^2 = 0.01$ and MAE being almost three times as high as the mean HS, see Figure 3). I am missing a discussion on potential applications where choosing SfM could be reasonable. Is SfM suited to measure shallow snowpacks? Maybe analyse your difference-maps also with regard to your topographic and soil features! For ii) a more thorough introduction and discussion of literature

working with snow distribution pattern is required (e.g. (Geissler et al., 2023; Pflug et al., 2021; Revuelto et al., 2020; Sturm & Wagner, 2010; Vögeli et al., 2016))

Regarding the suitableness of the SfM, we provided our answer above. Based on your suggestion regarding a more thorough literature review, we have expanded the discussion of snow distribution patterns by incorporating relevant studies that emphasize the importance of spatial variability in snow distribution under diverse environmental controls in the introduction as follows.

"For example, Sturm and Wagner (2010) found that snow depth patterns remain stable across years due to persistent topographic and vegetation influences in an Arctic region, highlighting the value of empirical snow distribution patterns for improving snow model accuracy. Vögeli et al. (2016) used high-resolution airborne digital sensors to refine precipitation scaling in a snow distribution model (Alpine3D), demonstrating the potential of remote sensing data to better simulate complex snow dynamics in alpine regions. Pflug et al. (2021) examined the interannual consistency of snow patterns and proposed a downscaling approach based on historical snow patterns in the California Tuolumne River Watershed, which is particularly useful for predicting snow distribution during years with limited observations. Revuelto et al. (2020) introduced a method combining in situ snow depth measurements with terrestrial laser scanner and time-lapse photography to produce temporal snow depth distribution patterns in a subalpine mountain environment, offering a transferable approach for deriving spatial snow data from limited ground observations."

**Minor Comments**

Title: Here and throughout the entire manuscript. I would avoid the word 'structures' in this context as snow structures could also refer to the microstructure (e.g. grain size, type or specific surface area) of the snow. Maybe use pattern or distribution instead.

Thank you for your suggestion. We replaced the word "structures" with "patterns" or "distributions" throughout the manuscript.

L15: LiDAR not introduced and make sure that you use the same abbreviations throughout your manuscript (lidar vs LiDAR).

The introduction of light detection and ranging (lidar) was added to the abstract and introduction and all uses of the term were updated to "lidar" for consistency.

L23: This sentence is not clear, you have only a very few measurements from the cameras and the measurements were taken at different locations. Thus, it is not surprising that they differ. I would skip this analysis.

Thank you for this feedback. Based on your earlier comment, we have removed the comparison of in situ techniques from the text.

L42: Repetition of L30ff.

The repetitive wording was revised.

L43: I would improve this section with a more thorough literature review. From what Geissler et al. (2023) and Pflug and Lundquist (2020) showed, patterns are rather stable.

Thank you for this feedback. We recently completed a stand-alone review that updated Clark et al. That will be incorporated in the introduction.

L56: …and most sensors cannot measure through the canopy of trees.

Indeed! We added this text to the sentence.

L64-65: The sensors can measure the HS, not the interactions between the snowpack and land/soil characteristics.

Agreed. Additional clarification has been added to the text:

"Hence, the capabilities of UAS platforms equipped with diverse sensors can observe snowpack properties and support analyses of field-scale physical interactions between snowpacks and land/soil characteristics (Cho et al., 2021)."

L70: Not sure what you mean with transition periods.

Additional clarification has been added to the text:

"However, investigating the transition periods between snow-on and snow-off poses challenges, primarily due to the snow becoming increasingly shallow and patchy, eventually revealing bare ground."

L76: Again, you mean spatial distribution and not structure of snow depth and probably persistence and not stability.

Structure was updated to distribution.

L88: Erroneous reference. Here and many others. Check entire manuscript.

This reference, as well as others throughout the manuscript, have been corrected.

L90: Actively managed and unmown grassland – sounds contradictory.

Agreed, we removed "and unmown".

L131: You need to specify the point densities of all of your surveys and data sets (Overall vs field vs forest) together with the size of the data gaps.

The point densities of the surveys and data sets were added to the supplementary materials.

L133: remove '-'

The dashes have been removed.

L147: How and when did you rasterize your products? Before or after the substruction from snow-off data? I am missing a co-registration of your datasets. See major comments.

Snow-on and snow-off DEMs (rasters) were made independently of each other. They were subtracted on a pixel-by-pixel basis to make each snow depth product. The methods description for both SfM and lidar now specify that the snow depths were calculated as the "difference between the ground classified snow-on and snow-off elevations within each pixel."

Co-registration was performed for both SfM and lidar DEMs based on the surveyed ground control points. Please see response to major comments for more information.

L167: 'made' – rephrase.

The sentence was rephrased as follows:

"Dense clouds were produced with the quality setting set to high and depth filtering set to moderate."

L171: Style again: …DEMs are derived based on the points classified as ground within…

This statement was rephrased as follows:

"Finally, digital elevation models (DEMs) were derived from the ground classified points within the dense clouds."

L182: © and TM antenna – please check author guidelines.

Thank you. We removed © and revised "Zephyr$^{TM\ antenna}$" to Zephyr antenna.

L194: Make sure to be consistent field-scale vs field scale.

Thank you for the comment. "field-scale" has been applied throughout the manuscript.

L196: Please clarify what variables are physical.

We clarified the text as follows.

"Land and soil characteristic variables are investigated as physical drivers of field-scale spatial distribution of snow depth. The variables used in this study are plant functional type, slope, aspect, shadow hours, saturated hydraulic conductivity (Ksat), and soil organic matter (SOM) (Figure 1)."

L201: Formula not referenced.

The formular was referenced in the text with GLI.

"The forested area was further classified as coniferous or deciduous for the study region by applying the Green Leaf Index (GLI) (Louhaichi, Borman, and Johnson 2001) (Equation 1) to the optical three-band (red, green, and blue) orthomosaics derived from the snow-off DJI Phantom 4 RTK survey."

$$GLI = \frac{(Green-Red)+(Green-Blue)}{(2*Green)+Red+Blue} \tag{1}$$

L211: At what date? The incidence angle changes throughout the season. And what about sub-canopy shadow hours? More details on the underlying method are needed! Check grammar.

Thank you for the thoughtful question. The current shadow hours were calculated on February 24$^{th}$. We agree that the change in solar angle between February 4 and March 7 could influence the shadow hours due to seasonal shifts in the sun's position, potentially impacting solar incidence and, thus, shadowing on the terrain. Although the sun's angle does change slightly during this period, our analysis assumed a consistent shadowing effect throughout the daytime for simplicity and practical applicability. The variation in solar angle over this period is relatively minor in terms of altering shadow duration or extent across the study area. Thus, we determined that incorporating adjustments in solar angles would have a

negligible effect on the overall shadow hours and snowpack conditions measured here. However, we acknowledge that incorporating dynamic solar angle adjustments could be beneficial in studies where seasonal or monthly changes in shadowing may significantly impact outcomes, particularly in cases involving longer observation windows or areas with high topographic relief.

To address the comment, we considered the shadow hours calculated using the unfiltered UAS LiDAR digital terrain model and a static sun incidence angle based on the average of February 4th and March 7th.

"The shadow hours were calculated using the unfiltered UAS LiDAR digital terrain model and a static sun incidence angle based on the average of February 4th and March 7th. Given the minor variation in solar angles between these dates, any change in shadow hours was considered negligible for this study."

L231: Something is wrong with the references.

Agreed. We modified this with the correct referring to Figure 2.

L234: Define winter season explicitly.

The sentence has been modified as follows. "Between December 15th, 2020 and March 12th, 2021, the average daily temperature was -2oC, the maximum daily temperature was 19°C on January 31st and the minimum daily temperature was -19°C on March 11th."

L236: I would rephrase this sentence.

The sentence has been rephrased "December and early January had ephemeral snowpacks of less than 10 cm which melted within a week. A snowpack was continuously present from late January through the middle of March."

L249: You have not really talked about standard deviations so far – this sentence is not clear? Is it needed at al?

The sentence has been removed.

L251: You did not introduce accumulation and ablation periods so far. What are you referring to? Sentence could become clearer after rephrasing.

The authors modified the sentence as follows. "Figures 2b and 2c show that the eight UAS-based SfM and lidar flights captured both the snow depth peaks and the ablation period following the last peak on February 20th, referring to the phase in the seasonal snow patterns when the snowpack begins to melt and decrease in depth."

L255: Could the increased variability of originate from the reduced point density/increased data gaps? -> Would become clearer with the transect plot (See major comments).

Please check our answer to the major comment with the supplemental figure and tables above.

L258: Remove 'snow observing'.

This has been removed.

L262-265: Unclear – sentences should be more concise.

Thank you for this comment. These lines (formerly L262 – 265) in section 4.1 were revised to make the discussion of the UAS vs in situ comparison clearer.

L271: r² - check layout!

This has been corrected.

L277: remove space in snow.

This has been corrected.

L287: 0 cm?

This statement was revised:

"The difference between SfM and lidar snow depths was fairly consistent in the field and close to 0 cm on most dates."

L291: remove daily

Removed as recommended.

L297 : 'were some gaps' – rephrase and be more precise. How do these gaps emerge? Using your formula 1, I assume that areas with no snow ('patchy snow cover') would result in a relative difference of -1. I think this needs to be further discussed as this is what makes your data set special and valuable to the community! There are not many data sets that have several revisits during one season and could be used to analyze also patchy snow covers and their evolution.

Thank you for highlighting this point. We agree that the presence of gaps in the snow cover warrants further clarification and discussion, especially given the uniqueness of our dataset. The gaps in snow cover in our study emerge due to several factors. First, variations in terrain, slope, and aspect affect snow accumulation and retention. Additionally, areas beneath dense canopy cover can experience faster snow ablation or limited initial accumulation, leading to patchy snow distribution across the field site.

Based on your suggestion, the statement has been rephrased "During the ablation period, consistent spatial patterns of the relative difference were observed despite the presence of patchy snow cover in some areas. These gaps emerged primarily due to differential melting, leading to sections with no remaining snow."

L299: over the time period – better: for all survey dates.

Replaced as recommended.

L303: AOI does not contain areas south of forests? And…

Agreed this is confusing. We were referring to the treeline at the northern part of our study error, not the forest. We have rephrased this to clarify the location and to make it less confusing.

L305: ..no forest is located in the wester field? This is very confusing. Maybe add small numbers that you could refer to Figure 7.

Agreed this is confusing. We modified the text so that we only refer to the forest area that is studied throughout. In this comment, we are referencing the forest. We are reviewing this section and may update the figure with numbers if the locations are not clear.

L311: You don't know what primarily drivers of snow distribution. This would require a more thorough analysis.

Agreed. We replaced this sentence with "In the combined areas (i.e., forest + field), relative snow depth clearly differs by vegetation type."

L330: It is well known that LiDAR outperforms SfM (Harder et al., 2020).

Additional text has been added to section 5.1 to discuss the relative performance of SfM and lidar. For example:

"While it is apparent that the accuracy of SfM-derived snow depth estimates cannot match that of lidar, the results of this study indicate that both techniques provide sufficient accuracy for monitoring the median change in shallow snow depths over time in flat, unforested land covers when there are a sufficient number of unique characteristics for SfM post processing. It is clear from the results of this study and previous ones that compared to in situ data, UAS lidar techniques produce lower errors and fewer data gaps than SfM, especially in forested land cover and over homogeneous snowpacks (Bühler et al., 2016; Bühler et al., 2017; Harder et al., 2020; Revuelto et al., 2021b; Miller et al., 2022)."

L337: What features? Can you give examples? This is where it gets interesting!

Thank you for this comment. The authors have revised section 5.1 to include more discussion on the relative performance of the two techniques and the limitations of SfM for the study region. We added more information on the greater potential for SfM tie point errors in forested/vegetated areas, over homogenous snowpacks (e.g., fresh snow), and in sub-optimal lighting conditions. Citations are presented for each of these conditions. In addition, further discussion was added to describe problem areas in the field at the Thompson Farm site (e.g., northwest field, east field) and potential causes for the larger differences in these areas (e.g., lack of unique features, fresh snow).

L340: Do you mean image overlap? You only have one point cloud for each survey.

This sentence has been rephrased to: "In SfM processing, an insufficient number of valid tie points, used to stitch together overlapping images, may degrade the accuracy of SfM snow depth data (Harder et al., 2016)."

L360: You only showed the overall relationships!

Modified the text to indicate that monitoring allows for identifying the overall relationships.

L375: Underneath rather than beneath?

The text was updated as suggested.

L386: would help the snow community.

The text was edited as suggested.

L388-389: Please give some examples for the 'numerous studies'. Not sure to what section/results you refer this time stability. This needs to be further analysed, see major comments.

References will be added to this section that indicate when snow depth patterns are consistent with or across seasons.

L411: remove space.

The space was removed.

**Tables and Figures:**

Table 1: These are the numbers of magnaprobe measurements per survey? Please clarify!

Upon further discussion and based on feedback from other reviewers, the authors determined that the inclusion of the full and partial sample dates together was likely to cause confusion and potentially bias the statistics computed for all dates. As a result, we have removed the extra samples collected on 2/4 and 2/24 and retained only those samples which were included in nearly all surveys. Table 1 was moved to the supplementary material (now Table S1) and the caption was revised to clarify what the numbers are referring to (i.e., number of locations where a grid of 9 samples was collected).

Figure 1: The patterns of the slope are interesting, as they vary on very different scales comparing forests and open sites. Can you confirm this from your knowledge of the sites or could this be due to some problems in the ground/no-ground classification of the point cloud? Here, again, a transect of the point clouds could help to get an idea of the topographic characteristics of your site. For the aspect: can you confirm these 'stripes' from your observations? Can you reproduce them with your SfM- snow-off map?

The slopes are indeed as they appear in this figure. The variations in the forest are likely why this land was not used as pasture. We are creating a transect that will show these patterns for the supplementary materials. Regarding aspect, some of the striping is due to the color range transition between 0 (red) and 360 (blue). This part of the field definitively has more "rows" as compared to the western and northwestern field; this is somewhat present in the orthomosaic.

Move coniferous into the middle of the dark green bar in the legend. Did you explain what these outlined areas are in the middle of the Western field and in between the western and eastern field?

Great point. We added text in the figure caption that indicates the outlined areas include a small pond, the USCRN station, a small outbuilding, and a section of dense shrubs that were not representative of the field.

Figure 2:

This plot is very unclear and it is very difficult to understand what you want to show. For instance, you can assign the uncertainty ranges to the individual dots. I would completely revise this Figure. Showing the meteorologic forcing together with the HS timeseries is important, but I would reorganize this figure. For instance, a plot containing Meteo forcings and the three (!) time series from your camera locations. A comparison with your available in-situ measurements is already done in Figure 3.

The caption has some formatting problems.

Thank you for your feedback on how to revise Figure 2. We have combined your feedback with that from the other reviewers and created a new version of this figure with three subplots. The purpose of this figure is to show the reader how typical conditions at the field site changed throughout the sampling period. We hope that the context provided by this time series also aids in the interpretation of the raster timeseries shown in figures 5 and 6.

The first subplot shows the meteorological data which illustrate the seasonal conditions (precipitation liquid equivalent and average air temperature). The second shows the snow depth evolution in the field from the field camera (continuous measurement) and the median snow depth value for the two UAS measurement types (SfM and Lidar). The third subplot shows the snow depth evolution in the forest from the two cameras and the median snow depth value for the two UAS measurement types. Field camera data was retained for this plot as it provides a continuous measurement of snow depths throughout the season.

[Figure]

**Figure 1 Time series of conditions at the Thompson Farm, Durham NH study area during the 2021 winter season including: hourly precipitation equivalent (mm) and temperature (°C) measured by a USCRN station (a) and daily camera snow depths and median UAS-measured snow depths in the field (b) and forest (c). Dates corresponding to the in situ and UAS sampling campaigns are marked by the dotted vertical lines. Periods where the temperature was colder than 0°C are indicated by the blue plot background and periods warmer than 0°C are indicated in pink in (a). Median SfM-measured snow depths in the forest on 2/4/21 and 2/28/21 exceeded 35 cm and are not shown (172 cm and 86 cm, respectively).**

Figure 3:

I would change the x-axis range to the data (e.g. 0 – 40 cm). I assume this plot combines all surveys? Clarify! It is hard to differentiate the colors of the SfM circles. After incorporating my major comments, it might be more suited to use the color of the dots to visualize different, more relevant informations such as the survey date instead of Magneprobe vs camera.

Yes, the plot combines all surveys. We have updated the figure caption accordingly. The authors have updated both of the subplots in this figure to a range from 0 – 50 cm on the x and y axes (this range matches the change made to figure 4). Based on your previous comment, the field camera data were removed from this figure and the updated version now compares the two UAS measurement techniques against the magnaprobe measurements. The colors and marker shapes have been updated to reflect this change.

[Figure]

**Figure 2 UAV-based snow depth measurements compared to in situ snow depth measurements from the magnaprobe. UAS-measured snow depths are shown for pixels overlapping the magnaprobe sample locations. Measurements collected on all dates are shown for the field (a) and forest (b). N-values indicate the number of samples shown within the plot axes. UAV-measured snow depths which exceeded 50 cm are not shown (1 SfM field, 6 SfM forest). Magnaprobe sample locations corresponding to UAS pixels with missing snow depth values are not shown (1 SfM field, 6 SfM forest, 7 lidar forest).**

Figure 4:

Is it needed to show all the outliers of your SfM data in this plot? I would be more interested in the Scatterplots and density plots for the more relevant range e.g. <50 cm.

Thank you for this suggestion. The plots have been updated to show a range from 0 to 50 cm. Based on feedback from other reviewers, the figure has also been divided into eight subplots showing the forest and field sections individually to improve visualization of this data.

[Figure]

**Figure 3 Comparison of SfM and lidar measurements for all sample dates by location. Scatterplots (left) compare snow depth for the three field areas (a, c, e) and the forest (g). Probability density plots (right) show the distribution of snow depth values for the three field areas (b, d, f) and the forest (h) by UAS technique. Measured snow depths exceeding 50 cm are not shown.**

Figure 5:

It seems to me as if you sometimes have less data gaps in the difference map compared to the SfM – how is that possible? (e.g. 2/20/21).

Thank you for your comment. The authors have reviewed the figures and determined that this was likely due to a data visualization issue. The snow depth plots only displayed values within the range shown in the legend (0 to 34 cm) and any values outside of this range were not included. However, in some locations the snow depths measured by SfM or Lidar were outside this range but were less than or equal to ±15 cm different from each other. To resolve this, the new figure was produced using filtered rasters which had their pixel values trimmed to the same range as the legend. The updated difference maps were calculated from these filtered rasters. As a result, the new versions of the difference maps also exclude these locations where snow depths exceeded reasonable values.

[Figure]

**Figure 4 Time series of snow depths for UAV Lidar and SfM in the field and forest. Difference is calculated as Lidar snow depth minus SfM snow depth. All values are shown in cm.**

Caption introduces SD. Either introduce the abbreviation for the entire manuscript or not.

Use of this abbreviation was removed from the caption.

Figure 7:

Are these dark-red areas along the edges of your study site potentially due to misclassifications (ground/no-ground) of your point cloud? Or can you explain them otherwise?

No, it is unlikely they are misclassified. As one approaches the northern edge of the field, there is a dense coniferous tree line. While we did not explicitly measure the processes in these locations, we were able to clearly see distinct patterns in snow depth. That said, during snowfall, we believe that interception seems to reduce the accumulation. There also seems to be a longwave effect where the warm trees melt the snow adjacent to the treeline. We are currently conducting other studies to better understand the energy balance in forest field transitions.

Figure 8: Why do you compare combined vs field and not field vs forest? Or all?

Thank you for this comment. We chose to compare combined vs. field areas, rather than field vs. forest or each area individually, because we assumed that plant functional types in forest areas primarily influence results, likely overshadowing the effects of other physical variables. However, we agree that including results for forest areas would benefit readers who may be interested. Please see the updated figure.

[Figure]

**Figure 5 Boxplots of the snow depth mean relative difference (MRD) by each physical feature (vegetation type, slope, shadow hours, Ksat, and soil organic matter) for (a) the combined areas (forest and field), (b) field, and (c) forest only. The**

**1-5 for each boxplot except for vegetation type represents the relative range of each physical variable in each area (For example, for slope in the combined areas, 1: 0-5 %, 2: 5-10%, 3: 10-15%, 4: 15-20%, and 5:20-25%).**

References

Geissler, J., Rathmann, L., & Weiler, M. (2023). Combining Daily Sensor Observations and Spatial LiDAR Data for Mapping Snow Water Equivalent in a Sub-Alpine Forest. *Water Resources Research*, *59*(9), Article e2023WR034460. https://doi.org/10.1029/2023WR034460

Harder, P., Pomeroy, J., & Helgason, W. D. (2020). Improving sub-canopy snow depth mapping with unmanned aerial vehicles: Lidar versus structure-from-motion techniques. *The Cryosphere*, *14*(6), 1919–1935. https://doi.org/10.5194/tc-14-1919-2020

Jennifer M. Jacobs, Adam G. Hunsaker, Franklin B. Sullivan, Michael Palace, & Eunsang Cho. (2020). *Shallow snow depth mapping with unmanned aerial systems lidar observations: A case study in Durham, New Hampshire, United States*.

Mazzotti, G., Webster, C., Quéno, L., Cluzet, B., & Jonas, T. (2023). Canopy structure, topography and weather are equally important drivers of small-scale snow cover dynamics in sub-alpine forests. *Hydrology and Earth System Sciences Discussions*, *2023*, 1–32. https://doi.org/10.5194/hess-2022-273

Pflug, J. M., Hughes, M., & Lundquist, J. D. (2021). Downscaling Snow Deposition Using Historic Snow Depth Patterns: Diagnosing Limitations From Snowfall Biases, Winter Snow Losses, and Interannual Snow Pattern Repeatability. *Water Resources Research*, *57*(8), Article e2021WR029999. https://doi.org/10.1029/2021WR029999

Pflug, J. M., & Lundquist, J. D. (2020). Inferring Distributed Snow Depth by Leveraging Snow Pattern Repeatability: Investigation Using 47 Lidar Observations in the Tuolumne Watershed, Sierra Nevada, California. *Water Resources Research*, *56*(9). https://doi.org/10.1029/2020WR027243

Revuelto, J., Alonso-González, E., & López-Moreno, J. I. (2020). Generation of daily high-spatial resolution snow depth maps from in-situ measurement and time-lapse photographs. *Cuadernos De Investigación Geográfica*, *46*(1), 59–79. https://doi.org/10.18172/cig.3801

Sturm, M., & Wagner, A. M. (2010). Using repeated patterns in snow distribution modeling: An Arctic example. *Water Resources Research*, *46*(12), Article 2010WR009434. https://doi.org/10.1029/2010WR009434

Vögeli, C., Lehning, M., Wever, N., & Bavay, M. (2016). Scaling Precipitation Input to Spatially Distributed Hydrological Models by Measured Snow Distribution. *Frontiers in Earth Science*, *4.* https://doi.org/10.3389/feart.2016.00108

Thank you for providing the important references. We incorporated them into our manuscript.

**Reviewer 2**

**Manuscript Title**: Characterizing Spatial Structures of Field-Scale Snowpack using Unpiloted Aerial System (UAS) Lidar and SfM Photogrammetry

**General review**: Cho and other authors measure the spatial heterogeneity of snow across a study plot in New Hampshire while evaluating the performance of UAS structure for motion (SfM) photogrammetry against lidar and in-situ observations during one snow season. This effort was conducted in New Hampshire, USA, and the study plot includes both a forested region and open field. The authors determined that the open areas tended to have deeper snow than forested areas and that lidar and SfM generally performed better, compared to in-situ observations and each other, in the open areas of the study plot. Static landscape variables, such as vegetation type and slope, impacted the distribution of snow consistently across the snow season. This project provides a new evaluation of UAS SfM as a tool to measure snow depth (a lower cost option compared to lidar). Much of the analysis, and thus manuscript text, includes a very clear description of the methodology used. However, the presentation of results in figures could be refined for clarity, and the results also require a more in-depth discussion around the performance of the select UAS tools, given the first two objectives of the study. The following line-by-line comments, which vary in "major" versus "minor" feedback, should provide more clarity and direction regarding this review, with the hope of better emphasizing the importance and value of this work.

[Answer] Thank you for your constructive feedback with specific comments on our manuscript. We have carefully revised our manuscript based on each of your comments.

**Line-by-line comments**

Abstract: Throughout, it is initially unclear and confusing what "spatial structure" is referring to. The words "patterns" and "spatial variability" are only used at the beginning of the introduction, which provide more clarity. Suggest briefly including a definition in the abstract, given the use of "spatial structures" in the manuscript title. Otherwise, suggest replacing "structure" with "variability, ""heterogeneity," or "distribution," which are more commonly used in the literature (including in the citations provided within this manuscript), whereas "structure" is often associated with the vertical microstructure of the snowpack.

We acknowledge the potential confusion around "spatial structure." The manuscript has been revised to clarify this term, either defining it briefly or substituting with "spatial variability" or "distribution" to align with conventional terminology.

Line 27-28: It would be beneficial and more complete to report, at the very least, the direction of the correlations.

Thank you for the suggestion. Since we did not explicitly conduct a collation analysis, we have revised the terminology and edited the text as follows.

"Within the field, the spatial distribution was primarily affected by slope and the shadowing effects of the forest canopy."

Line 36: Here "snowpack structure" is ambiguous, where, to some readership, the term insinuates the vertical, microscale structure of the snowpack.

The term has been replaced by "snowpack distribution".

Line 70: Unclear what "these transition periods" are referring to. Please define.

Additional clarification has been added to the text:

"However, investigating the transition periods between snow-on and snow-off poses challenges, primarily due to the snow becoming increasingly shallow and patchy, eventually revealing bare ground."

Line 73-75: It would be impactful for the authors to include why this type of forest/snowpack was chosen. For example, a number of the previously cited UAS works take place in other climates/forest types.

We conducted a review of Sturm and Liston's Climatology as well as Johnston et al.'s (2024) climatology and determined globally that ephemeral and transitional snowpacks cover large areas but are understudied. We have added text to reference the climatological information in Section 2.

"The snowpack at Thompson Farm is short-lived and warm, and snow climatologies from Sturm and Liston (2021) and Johnston et al. (2024) both classify the area as ephemeral. A review of existing research on the snow classes defined by Sturm and Liston (2021) and Johnston et al. (2024) determined that, despite covering large areas in the northern hemisphere, the ephemeral snow class is largely understudied, making new research on ephemeral snowpacks valuable."

Line 89: Reference error. There are a number of these throughout – also associated with figure references – thus I will only note this one.

This reference error, as well as others throughout the manuscript, have been corrected.

Section 2: Suggest further emphasis on why this area might be ideal for this type of study (shallow snow depths, type of forest, historical data, etc.). Obviously, there are many other locations which offer open versus forested regions.

In general, ephemeral and transitional snowpacks are understudied. The site has historical data and a rich history of forest ecology research.

Figure 1: It would potentially eliminate preemptive readership questions if the authors stated that the derivation of the variables shown in Figure 1b-g is explained in the following section (3).

Thank you for this suggestion. I agree that clarifying this in Figure 1 would be helpful for readers. We have added a note to the figure caption indicating that the derivation of variables in Figure 1b-g is explained in Section 3 to provide clearer guidance on where to find this information.

Table 1: The concept of a mixed forest is not also shown in Figure 1b. Is this referring to a blend of both coniferous and deciduous trees? If so, it is unclear the fraction of coniferous and deciduous used to determine the mixed forest area (50/50?).

Thank you for this feedback. Based on feedback from other reviewers, Table 1 was moved to the supplementary material (now Table S1) and now only includes sites which were visited during all of the

sampling campaigns. Extra samples collected on 2/4 and 2/24 were removed. The caption for Table S1 was updated to add further clarification of how these forest types were determined:

**Table S1** Number of magnaprobe sample locations (1 x 1 m grid cells) by land cover type for each snow-on UAV flight over the field campaign period in 2021. Each grid cell was comprised of nine, evenly spaced magnaprobe snow depth measurements and one snow tube snow water equivalent (SWE) measurement. Forest type was determined based on the binary Green Leaf Index (GLI) output within a 10 m x 10 m grid around the sample point (deciduous < 40% leaf on, mixed 40 – 60% leaf on, coniferous > 60% leaf on).

| Date | Number of sample locations (Field) | Number of sample locations (Coniferous) | Number of sample locations (Deciduous) | Number of sample locations (Mixed) |
|---|---|---|---|---|
| Feb 4[th] | 9 | 1 | 2 | 1 |
| Feb 10[th] | 9 | 0 | 3 | 1 |
| Feb 20[th] | 9 | 1 | 3 | 2 |
| Feb 23[th] | 9 | 1 | 3 | 2 |
| Feb 24[th] | 9 | 1 | 3 | 2 |
| Feb 28[th] | 9 | 1 | 3 | 2 |
| Mar 3[rd] | 8 | 1 | 2 | 2 |
| Mar 7[th] | 3 | 1 | 3 | 2 |

In addition, Figure S1 was added to the appendix to show the coarsened GLI output for these sites.

[Figure]

**Figure S1** Map of binary Green Leaf Index (GLI) 3 x 3 cm output (a) and 10 m x 10 m coarsened output (b) for study area at Thompson Farm in Durham, NH, USA.

Line 131: Can the authors provide insight on which ground conditions resulted in more returns versus less?

Thank you for your suggestion. Generally, ground conditions such as vegetation cover and surface roughness influenced the number of lidar returns. We applied this into the text.

Flights produced between a total of ~70-140 million returns per mission, depending on site ground conditions (e.g., vegetation cover, surface roughness, etc.).

Line 174: Suggest a Sturm citation for the magnaprobe.

Sturm, M. and Holmgren, J., 2018. An automatic snow depth probe for field validation campaigns. *Water Resources Research*, *54*(11), pp.9695-9701.

Line 180: As written, it is unclear if the 9 in-situ measurements were at random within the 1x1m or consistent across each survey. And can the authors please elaborate on why full sampling was not conducted during each flight (time/personnel constraints)?

Thank you for your comment. The 9 in-situ measurements were evenly spaced throughout the 1x1 m grid cell. Locations were relatively consistent, with small adjustments made between sample dates to avoid disturbed snow from previous sampling campaigns. The full sample campaigns were conducted on two of the dates as part of a prior study on comparison of in-situ snow depth measurement techniques (Proulx et al. 2023). Upon further discussion, the authors determined that the inclusion of both the full and partial sample data for this work was likely to cause confusion and potentially bias the statistics computed for each date. As a result, we removed the extra samples collected on 2/4 and 2/24 and retained only those samples which were included in nearly all surveys. Table 1 was moved to the supplementary material (now Table S1) and the caption was revised to specify what the numbers are referring to (i.e., number of locations where a grid of 9 samples were collected).

Additional text was added to section 3.2 to describe the sampling campaign and provide explanation for dates with missing samples.

"In-situ snow depth sampling was conducted in the field and forest using two methods: a Snow-Hydro LLC magnaprobe (Sturm and Holmgren, 2018) and three Moultrie Wingscapes Birdcam Pro Field Cameras. The magnaprobe sampling followed a single long transect (18 points) and two short transects (3 points each). The long transect was approximately 145 m long and laid out from east to west (Figure 1). From east to west, the transect started in the open field area, then transitioned to the coniferous, then mixed, and finally, deciduous forested areas. The two short transects were located in the open field; one in the northwest portion and the other in the southeast. At each point, nine, evenly spaced measurements were taken within 1 m x 1 m grid cells. It is worth noting that some dates were missing sample points due to disturbance of the sample area, either by collection on previous days or recreational use at the site, or due to personnel and equipment limitations (Table S1)."

Additional text was added to the Table S1 (formerly Table 1) caption in the supplementary material:

"Table S1. Number of magnaprobe sample locations (1x1 meter grid cells) by land cover type for each snow-on UAV flight over the field campaign period in 2021. Each grid cell was comprised of nine, evenly spaced Magnaprobe snow depth measurements and one snow tube SWE measurement. Forest type was

determined based on the binary GLI output within a 10x10m grid around the sample point (deciduous < 40% leaf on, mixed 40 – 60% leaf on, coniferous > 60% leaf on)."

Figure 2: It is currently challenging to determine the main takeaway of this figure – is it simply to observe the timeseries or to compare across the field versus forested areas? For example, it appears that air temperature and precipitation/cumulative precipitation are the same, which would make sense given data availability, but is thus redundant. It is particularly challenging to follow the 3x y-axis labels on the right side of each figure. Suggest reformatting as a sequence of timeseries – with only 1x precipitation/cumulative precipitation panel, 1x air temperature panel, and potentially 2x snow depth panels for each area (forest versus field), including the in-situ observations.

Thank you for your feedback on how to revise Figure 2. We have combined your feedback with that from the other reviewers and created a new version of this figure with three subplots. The purpose of this figure is to show the reader how typical conditions at the field site changed throughout the sampling period. We hope that the context provided by this time series also aids in the interpretation of the raster timeseries shown in figures 5 and 6.

To reduce redundant information, the first subplot shows the meteorological data which illustrates the seasonal conditions (precipitation liquid equivalent and average air temperature). Cumulative precipitation has been removed since the plot only shows the field season rather than the entire winter season. The second shows the snow depth evolution in the field from the field camera (continuous measurement) and the median snow depth value for the two UAS measurement types (SfM and Lidar). The third subplot shows the snow depth evolution in the forest from the two cameras and the median snow depth value for the two UAS measurement types.

[Figure]

**Figure 6** Time series of conditions at the Thompson Farm, Durham NH study area during the 2021 winter season including: hourly precipitation equivalent (mm) and temperature (°C) measured by a USCRN station (a) and daily camera snow depths and median UAS-measured snow depths in the field (b) and forest (c). Dates corresponding to the in situ and UAS sampling campaigns are marked by the dotted vertical lines. Periods where the temperature was colder than 0°C are indicated by the blue plot background and periods warmer than 0°C are indicated in pink in (a). Median SfM-measured snow depths in the forest on 2/4/21 and 2/28/21 exceeded 35 cm and are not shown (172 cm and 86 cm, respectively).

Figure 3: Are N-values the same across the two panels? Please add.

Thank you for this suggestion. N-values for the samples shown within the bounds of the plot axes have been added to each plot. The updated figure caption describes these N-values and also includes the number of points excluded from each plot.

Line 278-279: Suggest including the sub-areas of the field when introducing the field and an explanation as to why there the authors created a division here (e.g., what led to the decision making for a NW vs. W vs. E sub-area of the field?).

Thank you for this suggestion. The sub-areas of the field were selected based on a prior study at this site (Cho et al., 2021) and were not part of the original sampling design. As a result, the authors would prefer to introduce these field sections along with the results rather than in Figure 1. Additional information on the selection of these regions was added to section 4.2:

"The field was divided into three areas based on an early study (Cho et al., 2021) showing distinct topographic and soil characteristics in each section."

Figure 5: Suggest a more intuitive color scheme for snow depth. The difference color scheme makes sense (negative = red vs. positive = blue). For just snow depth, suggest purple leading to blue and then red (or something similar where red is not a color in the middle of the color bar).

Thank you for this suggestion. We have reviewed other journal articles which present spatial snow depth data measured by UAS and updated Figure 5 to a sequential, cool-toned color bar similar to that used elsewhere in the literature. The difference color scheme was kept nearly the same.

Figure 8: It is unclear what 1-5 (low to high) represents – this should be stated in the figure caption. Further, are there any statistical differences? If so, please note here and in the paragraph above with relevant p-values.

The 1-5 for each boxplot represents the relative range of each physical variable showing the histograms below (e.g., for slope, 1: 0-5 %, 2: 5-10%, 3: 10-15%, 4: 15-20%, and 5:20-25%). As suggested, they were specified in the figure caption.

To quantitively examine the statistical significances, the authors are analyzing them with a Kruskal-Wallis test, which is suitable for comparing medians across multiple groups and does not assume normal distribution. This test indicates that there are/are not statistically significant differences in MRD values among the physical variable groups. In response to your suggestion, we will also explore the use of notched boxplots as a visual indicator of median differences between groups. The notches provide an additional visual cue for significance; when notches between two groups do not overlap, this suggests a statistically significant difference between the medians at a 95% confidence level. These notched boxplots are now included in the revised Figure 8 to visually support the statistical findings.

Line 338: Here and throughout the manuscript, the terms "modestly" and "higher" read subjectively and would be more impactful if numerical values accompanied them and/or if there was a defined threshold for what the authors considered "modest" versus "high."

Agreed. The sentence has been revised to include numerical values as indicated below. Additionally, the authors have reviewed and revised similar subjective language throughout the manuscript.

"Compared to in-situ measurements, SfM experienced higher error in the field (MAD: 3.5 cm for lidar and 4.0 cm for SfM) and notably higher errors in the forest than lidar (MAD: 6.3 cm for lidar and 31.4 cm for SfM)"

Discussion: From the results section (e.g., Figure 4a, Figure 5), readers are led to believe that using SfM for snow depth generally is not a feasible option (without significant uncertainty) except for the west side of the field, and there isn't much of an explanation as to why. It is unclear what makes the west side of the field different from the rest of the field? Differences to the forested portion of the study plot are perhaps more obvious but are also not stated explicitly. Further, what might the authors suggest doing differently to reduce the numerous erroneous SfM measurements? The only plausible explanation currently provided is insufficient number of point clouds. It is stated in the introduction that this methodology is still an

emerging one, thus this seems like an opportunity provide insight into how UAS SfM for snow depth measurements may still evolve.

Thank you for this feedback. The authors reviewed the SfM rasters and workflow to determine if there was a possible explanation for the difference in performance in the western field. It appears that there were a greater number of "low confidence" tie points in the eastern field on 2/20 and 2/24, potentially contributing to the greater difference between techniques on these dates (Figure 5). New versions of figures 4 and 5 were produced using updated rasters which exclude these low confidence tie points. Low confidence points are likely a result of the fresh snow in this portion of the field lacking a sufficient number of unique characteristics for the SfM algorithms to stitch images together. This is also the likely cause of the large gaps in the SfM map on 2/4.

Options for improving the performance of SfM in the forest are limited due to the obstruction of the ground by branches and leaves. Mixed forests like the one in this study are also challenging due to the messy, repetitive features in the forest canopy which present challenges for finding valid tie points between images. While flying lower and slower can improve sensing in more open areas, lidar presents a much better alternative for measuring snow depths in forested regions.

While pixel-based comparison showed considerable scatter between the two techniques (Figure 4), both were able to capture the differences in snow depth over time (Figure 5). The authors have revised section 5.1 to include more discussion on the relative performance of the two techniques and the limitations of SfM for the study region. We added more information on the greater potential for SfM tie point errors in forested/vegetated areas, over homogenous snowpacks (e.g., fresh snow), and in sub-optimal lighting conditions. Citations are presented for each of these conditions. In addition, further discussion was added to describe problem areas in the field at the Thompson Farm site (e.g., northwest field, east field) and potential causes for the larger differences in these areas (e.g., lack of unique features, fresh snow).

Line 344: It would be helpful to connect the western portion of the field in this study to the subsequent sentences on past studies – e.g., does the western portion have a different vegetation type or other static landscape characteristic/combination of note (nothing particularly stood out in Figure 1)?

Thank you for this suggestion. We have added the following text to section 5.1:

"We also found that the SfM snow depths did not consistently agree with the lidar snow depths over the entire field and on all dates. On most dates, the difference in UAS-measured snow depths was close to 0 cm in the field. The best agreement occurred in the western portion of the field while the southeast and northwest portions had a larger amount of variability in measured values. The shadow hours and land cover type in the east and west fields are similar, however, the eastern field has a more gentle and less variable slope and fewer unique features (e.g., access road, USCRN station, pond, dirt piles, footprints) than the western field. The relatively homogenous features in the eastern field indicate that the difference between techniques is likely due to a lack of sufficient valid tie points for SfM. It is not clear what caused the differences between SfM and Lidar in the NW field that were not present in the other field areas. Unique features in the NW field are prevalent drainage patterns and shadowing that could be investigated further in the future."

Line 353: Can the authors indicate what likely caused the erroneous values of 150+ cm of snow depth as measured by SfM?

SfM point clouds are generally noisier than lidar because they are derived through calculation and not a direct measurement, however, the authors do not have an explanation for the exact cause of these erroneous values.

Conclusion: Suggest restating the error values when referring to "lower error"

The error values were restated for UAS techniques compared to in situ in the field and forest.

"Lidar demonstrated superior performance compared to SfM when evaluated against in-situ observations, exhibiting lower errors. Both UAS techniques exhibited lower errors in field settings (lidar MAD = 3.5 cm, SfM MAD = 4.0 cm) than in forested environments (lidar MAD = 6.3 cm, SfM MAD = 31.4 cm). Though, as expected, differences between lidar and SfM snow depths were more pronounced in forested regions (MAD = 55.7 cm), with SfM often registering anomalously deep snow depth values."

Line 431-433: Suggest explicitly restating the relationship – e.g., x vegetation type leads to deeper [or shallower] snow depth.

Thank you. We edit the statement below.

"The spatial distribution of snow depth captured by lidar remained consistent throughout the study period. For the entire study area, deeper snow was found in the field, in locations having shallow slopes and lower soil organic matter. Within the field, snow deepened with increasing shadow hours."
* * *
**Reviewer 3: Ross Palomaki**

**General Comments**

The authors present a generally well-written and well-referenced study comparing two UAS-based snow remote sensing techniques (Lidar and Structure from Motion) across different landscapes in New Hampshire, USA. I appreciate that the study was conducted over relatively shallow snowpacks (~tens of centimeters), which I feel are underrepresented in the literature and pose unique challenges for obtaining accurate snow depth measurements. I also like that the authors incorporated the Relative Difference concept to analyze this time series of spatially distributed snow depth measurements. I was not familiar with this concept, but I think it led to an interesting framing of the results.

While the introduction and methods sections are generally clear and easy to follow, there is room for improvement in the presentation of results and subsequent discussion. As noted in the Specific Comments below, I feel that the discussion is lacking critical engagement with some of the more complicated findings from this study. In particular, the SfM results do not inspire much confidence for the technique overall. I appreciate that the authors did not try to hide these larger errors, but there is little discussion around the potential sources of those errors, or suggestions for how these errors might be avoided in future studies (if that is even possible). This study could be more impactful to the broader snow/UAS community if some of these topics were explored more deeply in the discussion.

[Answer] Dr. Ross Palomaki, Thank you for your constructive feedback with specific suggestions on our manuscript. We have provided our answers to each of your comments and carefully revised our manuscript.

**Specific Comments**

Line 21: I am a bit surprised to see a range of errors for SfM but not lidar here, since they both have ranges on the next line. Perhaps an accidental omission?

Based on feedback from other reviewers, the comparison of in situ techniques was removed from the manuscript. As a result, the ranges were removed, and this section now only includes the MAE for the UAS and magnaprobe comparison.

Line 42-49: This paragraph would benefit from some additional discussion/more precise language related to spatial variability (Line 42), spatial patterns (Line 43), and hydrologic patterns (Line 44). With multiple phrases present it is unclear if these are different concepts, and which properties of the snow are relevant. E.g. if snow stratigraphy is variable over some length scale but snow depth is not, is that spatially variable snow but not a hydrologic pattern? Please clarify here and implement similar changes throughout the paper, e.g. Sections 4.3, 4.4, 5.

Thank you for pointing this out with reasonable suggestions.

The spatial variability of a snowpack encompasses both static (e.g., terrain, vegetation) and dynamic (e.g., weather, microclimate) factors that influence snowpack characteristics across different spatial scales (Clark et al., 2011; Grayson et al., 2002; Mott and Lehning, 2011; Trujillo et al., 2007). In this context, spatial variability refers broadly to the differences in snowpack properties (e.g., snow depth, density, temperature, and stratigraphy) across the landscape, which can vary from small scales (e.g., microtopographic variations) to larger watershed or landscape scales. By contrast, *spatial patterns* specifically denote the distribution of these snowpack properties as coherent structures or repeated features that emerge across the landscape. These patterns are critical as they represent consistent snowpack behavior linked to landscape features, reflecting hydrological, ecological, or even biological functions within the snowpack. When we refer to *hydrologic patterns*, we are primarily concerned with features of the snowpack that directly influence hydrological processes, such as snowmelt timing, runoff generation, or infiltration. These hydrologic patterns tend to persist over time but may be altered by major weather events, temperature changes, or snow redistribution by wind. For example, a hydrologic pattern might include consistent snow retention in areas with specific shading or wind exposure, directly affecting water availability during melt periods.

The spatial patterns of the snowpack, particularly those that exhibit consistency or repeatability over time, are instrumental in applications requiring snowpack assessment, such as snowmelt forecasting, remote sensing downscaling, in situ observation integration, and data assimilation for model enhancement (Pflug and Lundquist, 2020; Cho et al., 2023). Recognizing these patterns also aids in understanding landscape features, biogeochemical cycles, and habitat conditions (Boelman et al., 2019; Pflug et al., 2023).

By combining our answers with feedback from other reviewers, we implemented changes throughout the manuscript, particularly Sections 4.3, 4.4, and 4.5.

Line 55: Please quantify (at least approximately) "small-scale snow patterns" – tens of meters? It would also be helpful if you could relate this phrase to "field or local-scale snow features" at the beginning of this paragraph (Line 50).

We have modified the text to consistently use local and field scales and have defined the scales in the text.

Blöschl, G., Sivapalan, M., 1995. Scale issues in hydrological modelling: A review. Hydrological Processes 9, 251–290. https://doi.org/10.1002/hyp.3360090305

We use the Blöschl and Sivapalan (1995) definitions of scale where point or local (~1 m) and Hillslope or field (~100 m).

Line 69-70: "e.g., forest and fields" – I think more than just these landscapes. UAS technology has also enabled rapid progress in complex/mountainous terrain for example. I suggest removing this parenthetical statement from the end of the sentence.

The statement at the end of the sentence has been removed.

Line 70: "these transition periods" – unclear, previous sentence mentions "the entire snow period"

Additional clarification has been added to the text:

"However, investigating the transition periods between snow-on and snow-off poses challenges, primarily due to the snow becoming increasingly shallow and patchy, eventually revealing bare ground."

Line 75: Please clarify why you only investigate snow depths less than 35 cm. Is this just based on the datasets you collected or is there a particular motivation for snow depths below this threshold?

Thank you for the comment. We focused on snow depths less than 35 cm because this range reflects the specific conditions observed in our dataset.

We have updated the text for clarity, as follows:

(2) Conduct a quantitative comparison of lidar and SfM snow depths (< 35 cm) throughout the snow period, as this range of snow depth reflects the specific conditions observed in our dataset.

Figure 1: Consider widening the color bars in panels c-g. Panel b is wider and easier to discern the colors.

Agreed. We made the color bars widen in panels c-g.

Table 1 caption: The information about the sampling strategy in each 1x1 m grid cell seems better suited for the main text.

Based on feedback from other reviewers, Table 1 was moved to the supplementary material (now Table S1). The following text was also added to the main text (section 3.2) on Field Observations:

"At each point, nine, evenly spaced measurements were taken within 1 m x 1 m grid cells."

Line 149-152: With the in situ sampling strategy there are always 9 Magnaprobe measurements for the average snow depth. Is it possible to provide an approximate range of the number of lidar ground returns within each 1x1 m square? How does this number compare to the 9 Magnaprobe measurements, and is it relatively consistent throughout the study or does it change with time, landscape, etc?

A table summarizing the point counts by land cover for SfM and lidar was added to the supplementary material

Line 170: "following the same procedure as the lidar" – much work went into processing the lidar data. If you mean that you subtracted the snow-on and snow-off SfM maps to get snow depths, I suggest writing that explicitly here.

The statement was revised as follows:

"Snow depth products were derived following the same procedure as the lidar by calculating the difference between the ground classified snow-on and snow-off elevations within each pixel."

Line 180: Please clarify if the 9 measurements were taken in the same pattern at every grid cell ( I am envisioning a 3x3 pattern hitting all 4 corners and the middle of the cell, but this is worth specifying).

Thank you for your comment. The 9 in-situ measurements were evenly spaced throughout the 1x1 m grid cell. Locations were relatively consistent, with small adjustments made between sample dates to avoid disturbed snow from previous sampling campaigns. As we mentioned above, Table 1 was moved to the supplementary material (now Table S1) and the following text was also added to section 3.2 on Field Observations:

"At each point, nine, evenly spaced measurements were taken within 1 m x 1 m grid cells."

Line 183: does the less accurate GPS (~centimeter scale) matter compared to the RTK- driven image geotagging (listed as sub-centimeter in Line 155)?

Thank you for this comment. The use of "centimeter-scale accuracy" for the RTK image geotags was incorrect. The authors have updated the text to read:

"The RTK system integrates a static base station that relays GNSS corrections to the UAS, enabling approximately 3-cm accuracy of image geotags."

Line 210-211: Did you factor in a potential change in the shadow hours between February 4 and March 7 based on changing solar angles? Maybe this is negligible for the results of this study but would be good to clarify either way.

Thank you for the thoughtful question. The current shadow hours were calculated on February 24[th]. We agree that the change in solar angle between February 4 and March 7 could influence the shadow hours due to seasonal shifts in the sun's position, potentially impacting solar incidence and, thus, shadowing on the terrain. Although the sun's angle does change slightly during this period, our analysis assumed a consistent shadowing effect throughout the daytime for simplicity and practical applicability. The variation in solar angle over this period is relatively minor in terms of altering shadow duration or extent across the study area. Thus, we determined that incorporating adjustments in solar angles would have a negligible effect on the overall shadow hours and snowpack conditions measured here. However, we acknowledge that incorporating dynamic solar angle adjustments could be beneficial in studies where seasonal or monthly changes in shadowing may significantly impact outcomes, particularly in cases involving longer observation windows or areas with high topographic relief.

To address the concern, we considered the shadow hours calculated using the unfiltered UAS LiDAR digital terrain model and a static sun incidence angle based on the average of February 4th and March 7th.

"The shadow hours were calculated using the unfiltered UAS LiDAR digital terrain model and a static sun incidence angle based on the average of February 4th and March 7th. Given the minor variation in solar angles between these dates, any change in shadow hours was considered negligible for this study."

Line 230-249: Please clarify in this paragraph if USCRN precipitation values refer to snow depth or snow water equivalent. It would be helpful to specify in the first mention of the station (Lines 190-192) what sensor is available to measure precipitation and how you convert that to snow depth (assumed density?) if there is no dedicated snow depth sensor on the station.

Thank you for this comment. The precipitation depth from the CRN station represents a water equivalent (measured by weighing precipitation gage). The description of the CRN station at the end of section 3.2 was updated with this information and clarification was added to the Figure 2 caption.

Line 242-246: This is probably fine since we expect a fair amount of variation across the transects with a fairly shallow snowpack overall. But how did the field camera measurement compare with the closest 1-2 magnaprobe grid cells? I think that would provide more meaningful information than a comparison with the average across the entire transects. Similar comment for the forest site (Line 251-253).

Thank you for this feedback. Based on comments from other reviewers, the comparison between the in-situ measurement techniques was removed from the manuscript. As a result, this information was removed, and the text instead focuses on the comparison of UAS techniques.

Figure 2: I'm having a difficult time understanding what's going on here with 4 shared y- axes. Some points of confusion:

Cumulative precipitation doesn't start at 0, which implies it's cumulative from some date earlier than the start of the field campaigns. Maybe the start of the water year? But the field cameras at both sites imply 0 snow depth at the beginning of this timeseries. To me the cumulative precipitation does not provide any meaningful information in the context of these field campaigns. It also looks like the same curve in both subplots so I suggest at least removing one of the redundant curves, if not both.

I assume precipitation (mm) is in reference to snow water equivalent when the air temperatures are below 0 C. I suggest stating this explicitly somewhere. Is there a snow depth sensor on the USCRN station?

Caption states "UAS-based measurements represent average of all samples" but it also looks like some error bars are included, which are often covered up by different colored error bars from a different location. I suggest either removing the error bars completely and just stick to an average, or find some way to stagger the different sampling locations to prevent overlap. Also specify if the error bars represent IQR, +/- 1 standard deviation, etc.

One way to make this information clearer (with fewer shared axes) might be to have one subplot for temperature and precip (since the data are the same at both sites) and then separate subplots for the field and forest snow depth measurements.

Thank you for your feedback on how to revise Figure 2. We have combined your feedback with that from the other reviewers and created a new version of this figure with three subplots. The purpose of this figure

is to show the reader how typical conditions at the field site changed throughout the sampling period. We hope the context provided by this time series also aids in the interpretation of the raster timeseries shown in figures 5 and 6.

To reduce redundant information, the first subplot shows the meteorological data measured by the CRN station (precipitation liquid equivalent and average air temperature). Cumulative precipitation has been removed since the plot only shows the field season rather than the entire winter season. The precipitation depth from the CRN station represents a water equivalent (measured by weighing precipitation gage). The description of the CRN station at the end of section 3.2 was updated with this information and clarification was added to the Figure 2 caption.

The second subplot shows the snow depth evolution in the field from the field camera (continuous measurement) and the median snow depth value for the two UAS measurement types (SfM and Lidar). The third subplot shows the snow depth evolution in the forest from the two cameras and the median snow depth value for the two UAS measurement types. The authors feel that the error associated with these techniques is sufficiently described elsewhere in the manuscript (e.g., Figure 4) and so average values in Figure 2 are shown without error bars to improve visualization.

[Figure]

**Figure 7 Time series of conditions at the Thompson Farm, Durham NH study area during the 2021 winter season including: hourly precipitation equivalent (mm) and temperature (°C) measured by a USCRN station (a) and daily camera snow depths and median UAS-measured snow depths in the field (b) and forest (c). Dates corresponding to the in situ and UAS sampling campaigns are marked by the dotted vertical lines. Periods where the temperature was colder than 0°C are**

**indicated by the blue plot background and periods warmer than 0°C are indicated in pink in (a). Median SfM-measured snow depths in the forest on 2/4/21 and 2/28/21 exceeded 35 cm and are not shown (172 cm and 86 cm, respectively).**

Line 258-259: "All snow observing methods were able to distinguish that the average snow depth was slightly deeper in the forest than the field." Is this a mixup of forest and field?

Thank you for this comment. The text in section 4.1 were revised to better describe the updated figures. This statement was removed, and more detail was provided on the difference between measured snow depths in the field and forest.

Compare reported snow depths in Line 236-238 as well as results in Section 4.3.

We have updated section 4.1 to now include this comparison.

Figure 3: Please note in the caption the different axis limits between the subpanels.

Based on feedback from other reviewers, the authors have updated both of the subplots in this figure to a range from 0 – 50 cm on the x and y axes (this range matches the change made to figure 4). The field camera data was removed from this figure and the updated version now compares the two UAS measurement techniques against the magnaprobe measurements. The colors and marker shapes have been updated to reflect this change.

[Figure]

**Figure 8 UAV-based snow depth measurements compared to in situ snow depth measurements from the magnaprobe. UAS-measured snow depths are shown for pixels overlapping the magnaprobe sample locations. Measurements collected on all dates are shown for the field (a) and forest (b). N-values indicate the number of samples shown within the plot axes. UAV-measured snow depths which exceeded 50 cm are not shown (1 SfM field, 6 SfM forest). Magnaprobe sample locations corresponding to UAS pixels with missing snow depth values are not shown (1 SfM field, 6 SfM forest, 7 lidar forest).**

Figure 4: I like the subpanel showing the color coding of the different fields, but perhaps remove the forest outline as it took me a minute to realize those data are not included on the left figures. Also it is difficult to discern the difference between the solid Lidar lines and the dashed SfM lines in 4c.

Thank you for these suggestions. The forest outline has been removed from the reference map. Based on comments from other reviewers, we decided to reduce the x and y axes limits on these plots to 0 – 50 cm. The figure has been divided into eight subplots showing the forest and field sections individually to improve visualization of this data.

[Figure]

**Figure 9 Comparison of SfM and lidar measurements for all sample dates by location. Scatterplots (left) compare snow depth for the three field areas (a, c, e) and the forest (g). Probability density plots (right) show the distribution of snow depth values for the three field areas (b, d, f) and the forest (h) by UAS technique. Measured snow depths exceeding 50 cm are not shown.**

Figure 5: Personally I am not a big fan of the snow depth color bar and I'm not sure that it will be colorblind friendly. Did you try using a simple white -> blue gradient for snow depth? Your choice in the end, this is just a suggestion. The red -> blue gradient makes sense for the difference maps.

Thank you for this suggestion. We have reviewed other journal articles which present spatial snow depth data measured by UAS and updated Figure 5 to a sequential, cool-toned color bar similar to that used elsewhere in the literature. The difference color scheme was kept nearly the same.

[Figure]

**Figure 10 Time series of snow depths for UAV Lidar and SfM in the field and forest. Difference is calculated as Lidar snow depth minus SfM snow depth. All values are shown in cm.**

Figure 6/7: I suggest switching the order of these figures. The MRD map is a slightly easier concept for me to grasp and leads nicely into the individual RD maps. Plus it's nice to see the larger, detailed map before the smaller subpanels in the current Figure 6.

Thank you for this suggestion. We agreed on this point. The two figures were switched.

Figure 8: I really like this layout. Keeping five consistent boxplots is helpful across the different variables, and I appreciate the distributions below showing how they divide into the different boxplots. However, the discussion in Sections 4.4 and 5.2 would be strengthened if you could bring in some measure of statistical significance, e.g. Line 321- 323 "In the combined areas, the MRDs seem to decrease with increasing the Ksat values, except for the highest Ksat group, there are no significant patterns of MRDs when field areas are analyzed only." – how can you be certain there are no significant patterns

without a statistical test? Perhaps look into notched boxplots as a starting place, but there are other possibilities here.

Thank you for the insightful suggestion. We agree that incorporating statistical significance would strengthen the interpretations in Sections 4.4 and 5.2, particularly regarding trends in MRDs across the physical variable groups.

To address this, the authors analyzed them with a Kruskal-Wallis and Tukey test, which is suitable for comparing medians across multiple groups and does not assume normal distribution. This test indicates that there are/are not statistically significant differences in MRD values among the physical variable groups. We updated the text in Sections 4.4 and 5.2 to reflect the addition of these statistical tests and incorporated wording to clarify the observed trends in MRDs across the physical variable groups, based on both statistical testing and visual interpretation.

**Table S7**. Results of Kruskal-Wallis test (so-called "one-way ANOVA on ranks") for the mean relative difference of snow depth and physical variables for combined, field, and forest, respectively.

| variable | Combined | | Forest | | Field | |
|---|---|---|---|---|---|---|
| | Chi-squared | p-value | Chi-squared | p-value | Chi-squared | p-value |
| Slope | 23462 | 0.00** | 643 | 0.00** | 7 | 0.00** |
| Shadow hrs | 17346 | 0.00** | 83 | 0.00** | 7705 | 0.00** |
| Veg type | 29515 | 0.00** | 1293 | 0.00** | 2386 | 0.00** |
| K sat | 15781 | 0.00** | 641 | 0.00** | 91 | 0.00** |
| Soil Organic Matter | 17227 | 0.00** | 1540 | 0.00** | 2720 | 0.00** |

**Table S8**. Results of the Tukey test for the mean relative difference of snow depth with physical variables across all subgroups for combined, field, and forest, respectively.

| *Combined* | Slope | Shadow hrs | Veg type | Ksat | Soil Organic Matter |
|---|---|---|---|---|---|
| Group 1 vs. 2 | 0.00** | 0.00** | 0.00** | 0.079 | 0.00** |
| Group 1 vs. 3 | 0.00** | 0.00** | 0.00** | 0.00** | 0.00** |
| Group 1 vs. 4 | 0.00** | 0.00** | 0.00** | 0.00** | 0.00** |
| Group 1 vs. 5 | 0.00** | 0.00** | - | 0.675 | 0.00** |
| Group 2 vs. 3 | 0.00** | 0.00** | - | 0.00** | 0.00** |
| Group 2 vs. 4 | 0.00** | 0.00** | - | 0.00** | 0.00** |
| Group 2 vs. 5 | 0.00** | 0.00** | - | 0.264 | 0.00** |
| Group 3 vs. 4 | 0.00** | 0.00** | - | 0.00** | 0.00** |
| Group 3 vs. 5 | 0.052 | 0.00** | - | 0.00** | 0.00** |
| Group 4 vs. 5 | 0.539 | 0.00** | - | 0.00** | 0.00** |

| *Forest* | Slope | Shadow hrs | Veg type | Ksat | Soil Organic Matter |
|---|---|---|---|---|---|
| Group 1 vs. 2 | 0.00** | 0.44 | 0.00** | 0.00** | 0.00** |
| Group 1 vs. 3 | 0.00** | 0.34 | - | 0.00** | 0.00** |
| Group 1 vs. 4 | 0.00** | 0.99 | - | 0.00** | 0.00** |
| Group 1 vs. 5 | 0.00** | 0.00** | - | 0.00** | 0.00** |

| | | | - | | |
|---|---|---|---|---|---|
| Group 2 vs. 3 | 0.00** | 1.00 | - | 0.00** | 0.00** |
| Group 2 vs. 4 | 0.06 | 0.00** | - | 0.00** | 0.00** |
| Group 2 vs. 5 | 0.90 | 0.00** | - | 0.00** | 0.00** |
| Group 3 vs. 4 | 0.76 | 0.00** | - | 0.00** | 0.00** |
| Group 3 vs. 5 | 0.18 | 0.00** | - | 0.00** | 0.00** |
| Group 4 vs. 5 | 0.78 | 0.00** | - | 0.73 | 0.00** |

| *Field* | Slope | Shadow hrs | Ksat | Soil Organic Matter |
|---|---|---|---|---|
| Group 1 vs. 2 | 0.00** | 0.00** | 0.56 | 0.30 |
| Group 1 vs. 3 | 0.00** | 0.00** | 0.99 | 0.79 |
| Group 1 vs. 4 | 0.00** | 0.00** | 0.22 | 0.00** |
| Group 1 vs. 5 | 0.00** | 0.00** | 1.00 | 0.00** |
| Group 2 vs. 3 | 0.00** | 0.00** | 0.76 | 0.86 |
| Group 2 vs. 4 | 0.00** | 0.00** | 0.84 | 0.00** |
| Group 2 vs. 5 | 0.00** | 0.00** | 0.09 | 0.00** |
| Group 3 vs. 4 | 0.00** | 0.00** | 0.33 | 0.00** |
| Group 3 vs. 5 | 0.00** | 0.00** | 0.96 | 0.00** |
| Group 4 vs. 5 | 0.22 | 0.00** | 0.03 | 0.00** |

In Section 4.4 "To evaluate the effect of physical land characteristics on the spatial distribution of snow depth, the MRD values were analysed with respect to five land and soil characteristic values (e.g., vegetation type, slope, shadow hours, Ksat, and SOM) over the study domain. Boxplots of MRD by physical feature are shown for the combined forest and field areas, field only, and forest only (Figure 8). Statistical significance results among groups, based on Kruskal-Wallis and Tukey tests summarized in Tables S7 and S8. In the combined areas (i.e., forest + field), relative snow depth significantly differs by vegetation type. Coniferous forests have low MRDs (mean: -0.36) which indicates that snow in those areas is shallower relative to the spatial mean of snow depth by around 36%. For the deciduous forest, the mean MRD is -0.2 with a wide interquartile range from –0.23 to 0.19. MRD values in the field are higher compared to the two forest types which ranged from –0.11 to 0.22 (mean: 0.08). For the combined areas as well as field and forest only, slope contributes to snowpack spatial patterns, even though the study area has a gentle slope (less than 20%). High MRDs are found in flat areas (0 – 5% slope) and gradually decrease with increasing slope. The effect of slope for the forest only area is relatively modest. The shadow hours show a clear but contradictory contribution to snow depth patterns in the field and forest only areas as compared to the combined area. When the field and forest are separated out, low MRDs are found in areas where shadow hours are short (e.g., less than 2 hours), and the MRDs gradually increase with increasing shadow hours. For the combined area, the highest shadow hours had the lowest snow depth, but this is likely the result of a mixed effect due to the dense shading in the coniferous forest. Ksat shows little evidence of contributing to the spatial distribution of snow depth in the field, but there are distinct differences in MRDs among lower Ksat groups in the forest. In the combined areas, MRDs tend to decrease with increasing the Ksat values, except for the highest Ksat group. Compared to Ksat, SOM exhibits a clearer pattern with MRD decreasing as SOM increases in both the combined areas and field analysis. The forest area does not display consistent MRD patterns with changes in SOM."

In Section 5.2 "As compared to vegetation and terrain characteristics, few studies have examined the influence of soil characteristics on the snowpack. Our results indicate that snowpack depth decreases statistically significantly with increasing SOM (significant level < 0.01). This finding aligns with our previous study, which utilized maximum entropy modeling to analyze spatial variations of shallow snowpack over the same domain but during different periods (Cho et al., 2021). Even though a clear relationship between Ksat and snowpack was not found in this study (Figure 8b), it is acknowledged that soil thermal properties, such as the thermal conductivity of the soil underneath the snowpack, generally influence the rate of heat transfer between the snow and soil layers (Kane et al., 2001; Zhang, 2005). Also, the moisture content of the soil can affect the distribution of soil frost (Bay et al., 1952) and snowpack because the energy transfer at the snow-soil interface is controlled by wetness of the soil (Bay et al., 1952; Fu et al., 2018). Although spatial distribution of soil moisture is typically considered to be constant (frozen) during winter, intermediate rainfall events and freeze-thaw cycles can dramatically change the spatial patterns of soil moisture and freeze-thaw states in regions having ephemeral snowpacks. This can be important because the thermal conductivity in frozen state is more sensitive to soil type than non-frozen condition, because the thermal conductivity of ice is four times larger than that of liquid phase (Penner, 1970)."

Section 5.1: To me this discussion is lacking critical engagement with some of the more complicated findings from this study. I'm not sure I agree that "It is clear from the results of this study and previous ones that both UAS SfM and lidar techniques provide a viable method for monitoring snow depth change across many land cover types." (line 358- 359) based on the SfM results in Figures 3 and 4 where the SfM depths are anywhere from 2-10 times larger than the in situ measurements. I doubt there are many applications where errors of this magnitude are acceptable. Additionally it doesn't seem feasible to rely on the SfM technique in forested areas based on all the missing data in Figure 4. Can you expand upon either of these? You briefly mention overcast skies possibly affecting SfM data collection (line 337) but this doesn't seem to explain why the SfM snow depths in the western field had much better agreement than the E and NW fields (Figure 4). What was the vegetation like in the fields? Was it fully buried by snow or partially extending above the snowpack? Could there be GPS/processing errors affecting the final results? Including individual photos from the SfM photosets could help illustrate some of the challenges.

Thank you for this feedback. The authors agree that the statement on line 358 – 359 did not accurately represent the findings of this study. We have updated the statement as follows:

"While it is apparent that the accuracy of SfM-derived snow depth estimates cannot match that of lidar, the results of this study indicate that both techniques provide sufficient accuracy for monitoring the median change in shallow snow depths over time in flat, unforested land covers when there are a sufficient number of unique characteristics for SfM post processing. It is clear from the results of this study and previous ones that compared to in situ data, UAS lidar techniques produce lower errors and fewer data gaps than SfM, especially in forested land cover and over homogeneous snowpacks (Bühler et al., 2016; Bühler et al., 2017; Harder et al., 2020; Revuelto et al., 2021b; Miller et al., 2022)."

Additional discussion on the limitations of SfM in the forest and over homogeneous snowpacks was also added to section 5.1. The authors reviewed the SfM rasters and workflow to determine if there was a possible explanation for the difference in performance in the western field. It appears that there were a greater number of "low confidence" tie points in the eastern field on 2/20 and 2/24, potentially contributing to the greater difference between techniques on these dates (Figure 5). New versions of

figures 4 and 5 were produced using updated rasters which exclude these low confidence tie points. Low confidence points are likely a result of the fresh snow in this portion of the field lacking a sufficient number of unique characteristics for the SfM algorithms to stitch images together. This is also the likely cause of the large gaps in the field portion of the SfM map on 2/4.

The authors have revised section 5.1 to include more discussion on the relative performance of the two techniques and the limitations of SfM for the study region. We added more information on the greater potential for SfM tie point errors in forested/vegetated areas, over homogenous snowpacks (e.g., fresh snow), and in sub-optimal lighting conditions. Citations are presented for each of these conditions. In addition, further discussion was added to describe problem areas in the field at the Thompson Farm site (e.g., northwest field, east field) and potential causes for the larger differences in these areas (e.g., lack of unique features, fresh snow).

Section 5.2: Similar to a comment above, this section would be stronger if the relationships between physical variables and snow depth could be quantified statistically.

Thank you. We have updated this section to reflect the addition of the statistical tests to statistically quantify the observed trends in MRDs across the physical variable groups.

Lines 409-422: In the description of the in situ data collection you noted that one SWE sample was collected in each grid cell. Did you try any analysis with those measurements?

We determined the density and SWE and briefly looked at differences between the field and forest. However, because the snowpack was quite shallow, we believe that there was considerable error in those measurements and are not comfortable using them for research purposes.

**Technical Corrections**

Line 26: all areas → both areas? Complete

Line 88-89: Missing reference Corrected

Line 122: acronym IR not defined Corrected

Line 154: acronym CMOS not defined Complete

Line 182: remove superscript formatting from "antenna" Complete

Line 185-186: Typo in personal communication date? Data for this study collected in 2021 but personal communication listed as 2023 Corrected to 2020

Line 200: Missing reference Corrected

Line 200-201: "snow-off" Corrected

Line 231: Missing figure number Corrected

Line 242: Missing figure number Corrected

Line 277: Missing reference Corrected

Line 283: Missing reference Corrected

---

## Referee Report (RR1)

**General Feedback:**

Overall, the authors have carefully revised the manuscript, incorporating the feedback from the reviewers, which is greatly appreciated. The result is an improved manuscript that, along with the dataset and results, will make a valuable contribution. However, I have a few comments that I would like to see addressed before final approval:

**Major Comments**

The co-registration of your SfM and LiDAR products is still unclear. In your answer, you say that you used Ground Control Points to do so:

"All of the surveys were co-registered using ground control points(..). Linear, horizontal, and vertical shifts were applied to align all digital elevation models to these GCPs."

In the text you say that:

"GCPs surveyed using the base/rover equipment were used to co-register the UAS data. Linear, horizontal, and vertical shifts were applied to align all SfM and lidar DEMs to the GCPs."

I would be interested in what GCPs were used? If only optical GCPs were used, how were they identified in the LiDAR point clouds? What were the magnitudes of applied shifts for sfm vs lidar? I think this could be another advantage of LiDAR sensors that could be discussed with your data: Were the GCPs really needed for the LiDAR data? What is the benefit of the additional effort?

**Minor Comments**

L16: Add a sentence on why a better understanding is needed for your specific environment.

L23: Avoid saying 0 cm.

L24: Remove "also"

L39: Add examples to static and dynamic fluxes/variables

L47: Numerous... various.. This sentence needs to be revised.

L78: This reasoning is not very convincing. "Growing need for understanding of UAS sensor's strengths and weaknesses". I would agree, but I don't see the link to the next sentence: "However, it is challenging to measure shallow snowpacks". As this is the key motivation of this work, I suggest rephrasing. Suggestion: 1) Need for new, multi-temporal data sets. 2) This is specifically the case for transition periods and shallow snowpacks (consider citing https://doi.org/10.1016/j.earscirev.2024.104751) 3) Various sensors exists with strengths and weaknesses that need to be investigated for your specific hardware and environment.

L91: are discussed in Sections 4.3 and 4.4.... this should be your result section!

L110: Great! I missed that in your introduction/motivation. Maybe add a sentence on: https://doi.org/10.1016/j.earscirev.2024.104751

L270: (Figure 3)

Section 4.3: I found it confusing to see Figure 6 ("Mean relative difference" – exploring the snow distribution) right after Figure 5 ("lidar and sfm difference"- exploring the system differences). It was not directly clear what difference you are talking about in this section. Maybe you could use more easy to follow acronyms or write one additional introduction sentence in this section.

L335: Not easy to follow. I can see in Figure 5 that relative difference maps are similar during accumulation period. During ablation (March?), I can see more white areas, but not the "consistent spatial patterns" that you are talking about. Maybe add some numbers/letters to guide the reader to the individual features you are talking about/comparing? Are the white areas areas with no snow? Please add to the caption/legend what white areas are.

L398ff: Are these your values or the literature values? I suggest providing your RMSE values and the ranges suggested by the literature.

L432: Similar to comment above, I would recommend to be more precise: "By comparing maps of snow depth change…" You compared MRD not maps of snow depth change, right? Shouldn't this paragraph be included into the next section?

L440: This first sentence could be repharsed.

Figure S2: Legend and scale bar are missing.

---

## Author Response (AR2)

**Editor Comment**

Dear authors,
the paper was again reviewed by 2 reviewers - both are in general happy with the new version of the manuscript. However, a couple of comments by reviewer #1 need to be considered before I can consider the manuscript to be published in HESS. Please make sure that you carefully address these points and provide a new version with changes marked so I can easily see what you did and make a decision based on your revised version.
Best regards
Markus Weiler

We sincerely thank you for your continued evaluation of our revised manuscript. We appreciate the opportunity to further improve our work and believe the manuscript has benefited significantly from this round of feedback. Below, we provide detailed responses to each of the remaining points, with changes clearly marked in the revised manuscript.

**Reviewer #1**

**General Feedback:**

Overall, the authors have carefully revised the manuscript, incorporating the feedback from the reviewers, which is greatly appreciated. The result is an improved manuscript that, along with the dataset and results, will make a valuable contribution. However, I have a few comments that I would like to see addressed before final approval:

Thank you again for your constructive feedback with specific comments on our manuscript. We have carefully revised our manuscript based on each of your comments.

**Major Comments**

The co-registration of your SfM and LiDAR products is still unclear. In your answer, you say that you used Ground Control Points to do so:

"All of the surveys were co-registered using ground control points(..). Linear, horizontal, and vertical shifts were applied to align all digital elevation models to these GCPs."

In the text you say that:

"GCPs surveyed using the base/rover equipment were used to co-register the UAS data. Linear, horizontal, and vertical shifts were applied to align all SfM and lidar DEMs to the GCPs."

I would be interested in what GCPs were used? If only optical GCPs were used, how were they identified in the LiDAR point clouds? What were the magnitudes of applied shifts for sfm vs lidar? I think this could be another advantage of LiDAR sensors that could be discussed with your data: Were the GCPs really needed for the LiDAR data? What is the benefit of the additional effort?

**Response:**
Thank you for this important clarification request. Conventional optical GCPs were surveyed-in using ground based GNSS RTK equipment prior to each successive UAS lidar and optical flight. UAS lidar trajectories and ground based GNSS positions were corrected using the same GNSS base station. Because of system limitations with our optical UAS, a separate base station had to be used to apply RTK corrections to the UAS imagery geotags. A constant linear shift was therefore apparent when comparing the GCP positions in the optical orthomosaics to the measured positions, as the base station used to correct the UAS imagery geotags was not surveyed-in to a high degree of accuracy. A total linear shift of ~2.5m was generally required to ensure that the photogrammetry products aligned with Lidar products.

Although UAS lidar data often require fewer GCPs due to higher onboard geolocation accuracy, we chose to use the same set of GCPs across both systems to maintain consistency and ensure the comparability of SfM and lidar products. This additional effort provided a common geospatial reference for both datasets and improved the reliability of relative difference analyses between sensor types.

**Minor Comments**

L16: Add a sentence on why a better understanding is needed for your specific environment.

**Response:** We added a sentence to the introduction to clarify the relevance of our study environment:
*"This is particularly critical in mixed vegetation environments like ours, where both forest canopy and open areas influence snow accumulation and melt patterns."*

L23: Avoid saying 0 cm.

To avoid that, we revised the sentence as below.
*"Snow depth maps from SfM and lidar were fairly consistent in the field, with only marginal differences on most dates."*

L24: Remove "also"

"Also" has been removed.

L39: Add examples to static and dynamic fluxes/variables

**Response:** We now state:
*"The spatial variability of a snowpack is a function of static (e.g., slope, aspect, vegetation type, soil properties) and dynamic variables (e.g., solar radiation, wind direction and speed, temperature) and fluxes over a range of spatial scales"*

L47: Numerous… various.. This sentence needs to be revised.

**Response**: Rephrased for clarity and conciseness.

*"Previous studies have proposed diverse approaches to characterize snow distribution patterns and their temporal evolution across a range of climatic and topographic settings."*

L78: This reasoning is not very convincing. "Growing need for understanding of UAS sensor's strengths and weaknesses". I would agree, but I don't see the link to the next sentence: "However, it is challenging to measure shallow snowpacks". As this is the key motivation of this work, I suggest rephrasing. Suggestion: 1) Need for new, multi-temporal data sets. 2) This is specifically the case for transition periods and shallow snowpacks (consider citing https://doi.org/10.1016/j.earscirev.2024.104751) 3) Various sensors exists with strengths and weaknesses that need to be investigated for your specific hardware and environment.

**Response:** Thank you for your suggestion. We revised this paragraph using your suggested structure and cited the recommended paper (López-Moreno et al., 2024):
*"There is a growing need for new, multi-temporal snow datasets, especially during transition periods such as shallow and patchy snow conditions. These periods pose measurement challenges but are key for understanding snowpack dynamics (López-Moreno et al., 2024). Different UAS-based sensors offer complementary strengths and weaknesses that warrant further investigation in various environments."*

L91: are discussed in Sections 4.3 and 4.4…. this should be your result section!

**Response**: To clearly say, the sentence has been revised as below.

*"Sections 4.3 and 4.4 further examine the spatial patterns and temporal dynamics of snow depth, along with the physical variables influencing these patterns."*

L110: Great! I missed that in your introduction/motivation. Maybe add a sentence on: https://doi.org/10.1016/j.earscirev.2024.104751

**Response**: Added a sentence here.

*"López-Moreno et al. (2024) also emphasized the importance of studying these transient snow conditions, highlighting their sensitivity to climate variability and their implications for hydrological and ecological processes."*

L270: (Figure 3) Section 4.3: I found it confusing to see Figure 6 ("Mean relative difference" – exploring the snow distribution) right after Figure 5 ("lidar and sfm difference"- exploring the system differences). It was not directly clear what difference you are talking about in this section. Maybe you could use more easy to follow acronyms or write one additional introduction sentence in this section.

**Response:** We have added an introductory sentence to Section 4.3 to provide clearer context for the subsequent analysis of MRD patterns.

"Understanding and quantifying the spatio-temporal variability—or stability—of snowpack is essential for identifying the physical drivers that influence snow accumulation and ablation

across heterogeneous landscapes. To explore these dynamics in detail, MRD values were mapped to reveal spatial patterns in snow depth across survey dates (Figure 6)."

L335: Not easy to follow. I can see in Figure 5 that relative difference maps are similar during accumulation period. During ablation (March?), I can see more white areas, but not the "consistent spatial patterns" that you are talking about. Maybe add some numbers/letters to guide the reader to the individual features you are talking about/comparing? Are the white areas with no snow? Please add to the caption/legend what white areas are.

**Response:** To aid reader interpretation, we included more detailed descriptions referencing specific regions (e.g., northern/eastern field and forest areas). Additionally, figure captions now clarify that white areas represent locations with no snow cover (i.e., bare ground).

*During the ablation period, consistent spatial patterns of the relative difference were still observed, particularly in the northern/eastern field where snow remained relatively deep, in contrast to the forest areas which continued to show shallow snow or exposed ground. These patterns persisted despite the increased presence of patchy snow cover in some regions.*

*Caption: Figure 7 Relative difference maps generated from the UAS lidar-based snow depth maps from February 4th to March 7th. The white areas in the figures indicate either masked areas (e.g., ponds and facilities) or areas with no snow.*

L398ff: Are these your values or the literature values? I suggest providing your RMSE values and the ranges suggested by the literature.

**Response:** Thank you for the suggestion. We have clarified that the reported values include both our observed RMSD results and the ranges from previous literature.

*SfM-derived error values from our study were 4.0 cm MAD and 6.8 cm RMSD for the field, and 31 cm MAD and 71 cm RMSD for forested areas, highlighting a clear vegetation-dependent variation in accuracy. These findings are consistent with previous studies comparing UAS SfM and snow probe measurements, which report RMSD values typically below 31 cm in sparsely vegetated or alpine environments, increasing up to 37 cm in areas with denser vegetation such as bushes, tall grass, or forests (De Michele et al., 2016; Bühler et al., 2016; Avanzi et al., 2018; Belmonte et al., 2021).*

L432: Similar to comment above, I would recommend to be more precise: "By comparing maps of snow depth change…" You compared MRD not maps of snow depth change, right? Shouldn't this paragraph be included into the next section?

**Response:** Thank you for the clarification. We revised the sentence to accurately reflect the analysis performed. The updated sentence now reads:
*"By comparing maps of mean relative difference (MRD) with maps of physical variables at the site, specific factors influencing snowpack dynamics over the winter season were identified."*

The paragraph was combined to the following section for better alignment.

*"With limited wind redistribution in the study area, the time stability analysis indicated that relative differences in the snowpack were generally consistent throughout both the accumulation and ablation periods.In addition to the previous findings that snowpack patterns are relatively consistent from year to year (Pflug and Lundquist, 2020; Revuelto et al., 2014), this study demonstrates that fixed physical variables such as vegetation, topography, and soil characteristics can sufficiently control the spatial variations of snowpack throughout a winter period. By comparing maps of mean relative difference (MRD) with maps of physical variables at the site, specific factors influencing snowpack dynamics over the winter season were identified. Our findings highlighted that vegetation type is a dominant factor shaping snow depth patterns. In both combined and field-only areas, SOM showed a statistically significant relationship, with snow depth decreasing as SOM increased. Furthermore, shadow hours and slope were found to contribute to the spatial variability of snowpack, even though the study area features relatively gentle slopes. The findings regarding the influence of vegetation and topographical factors on the snowpack's spatial variability align with previous studies conducted (Currier and Lundquist, 2018; Deems et al., 2006; Trujillo et al., 2007)."*

L440: This first sentence could be rephrased.

**Response**: Sentence has been revised for clarity and flow.

*"With limited wind redistribution in the study area, the time stability analysis indicated that relative differences in the snowpack were generally consistent throughout both the accumulation and ablation periods."*

Figure S2: Legend and scale bar are missing

**Response:** We have updated Figure S2 in the Supplementary Material to include both a legend and scale bar.

---

## Author Response (AR3)

**Editor comment**

Dear authors,
I am happy with the revised manuscript and so we can publish it in HESS. However, you need to change the author list as indicated "Please note that the sign "†" is usually used to mark up the deceased authors."
Best regards
Markus

**Answer:**

Thank you for your guidance regarding the use of symbols in the author list. To align with Copernicus Publications' conventions, I will replace the dagger symbol (†) with an asterisk (*) to denote the author's equal contribution.